

 SciPost Phys. Lect. Notes 100 (2025)

# Numerical aspects of large deviations

**Alexander K. Hartmann**

Institute of Physics, University of Oldenburg, Germany

a.hartmann@uni-oldenburg.de

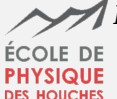

*Part of the 2024-07: Theory of Large Deviations and Applications collection*
*Session 123 of the Les Houches School, July 2024*
*published in the Les Houches Summer School Lecture Notes series*

## Abstract

**An introduction to numerical large-deviation sampling is provided. First, direct biasing with a known distribution is explained. As simple example, the Bernoulli process is used throughout the text. Next, Markov chain Monte Carlo (MCMC) simulations are introduced. In particular, the Metropolis-Hastings algorithm is explained. As first implementation of MCMC, sampling of the plain Bernoulli model is shown. Next, an exponential bias is used for the same model, which allows one to obtain the tails of the distribution of a measurable quantity. This approach is generalized to MCMC simulations, where the states are vectors of $U(0,1)$ random entries. This allows one to use the exponential or any other bias to access the large-deviation properties of rather arbitrary random processes. Finally, some recent research applications to study more complex models are discussed.**

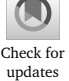

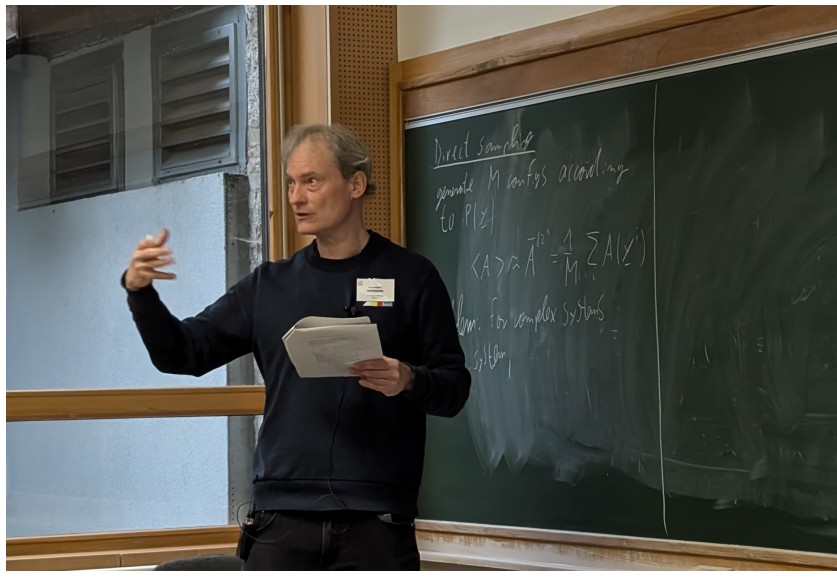

## 1 Motivation

The aim of theoretical physics is to understand nature by providing models, which are as simple as possible to explain the targeted phenomena. Unfortunately, most models cannot be solved analytically. Thus, one has to use numerical simulations [1] to investigate them. Here, models for stochastic processes are considered, which means that quantities of interest are random variables and characterized by probability distributions. In this work, basic methods are introduced, which allow one to obtain the probability distribution of the quantity of interest over a large range of the support, down to very small probabilities.

As toy example for the following content, we use a very simple model, i.e., the Bernoulli process. This is a series of $n$ independent coin flips with outcomes $y_i \in \{0, 1\}$, $\underline{y} = (y_1, \ldots, y_n)$. Let $\alpha$ be the probability of obtaining the result 1 in a single coin flip, i.e.

$$P(y_i = 1) = \alpha, \qquad P(y_i = 0) = 1 - \alpha. \tag{1}$$

Thus, the probability for a certain outcome $\underline{y}$ is

$$P(\underline{y}) = \prod_{i=1}^{n} \alpha^{y_i}(1-\alpha)^{1-y_i} = \alpha^{l(\underline{y})}(1-\alpha)^{n-l(\underline{y})}, \tag{2}$$

where $l = l(\underline{y}) = \sum_i y_i$ is the number of 1's in the $n$-coin experiment. Thus, $l$ is independent of the actual order of 0's and 1's in $\underline{y}$. Here, we will be mainly interested in the distribution of $l$. It is well known that $l$ is distributed according to the Binomial distribution. Thus, the probability to find for $l(\underline{y})$ a specific value $l \in \{0, \ldots, n\}$ is given by

$$P_n(l) = \binom{n}{l}\alpha^l(1-\alpha)^{n-l}. \tag{3}$$

This allows for a simple comparison of the numerical results with the known distribution.

The present text tries to be very comprehensive, including also very fundamental material, which many readers might know partially. Thus, a short overview is given now, to allow the reader maybe to skip some sections. In section two, *simple sampling* is introduced, which means that a probability distribution is sampled according its original underlying distribution. In the best case, this is done by *direct sampling*, where each call in the program to a corresponding function yields an independent realization of the stochastic process. This is applied to the Bernoulli process. Then *biased sampling* is introduced, which means that the system is sampled according to a distribution different from the original one. First, the *educated* variant is discussed, where one knows how one has to "push" a system by changing system parameters to sample it in the region of interest. Again the Bernoulli model is considered in test simulations, which shows than indeed the tails of the distributions can be reached.

In the third section, *Markov chain Monte Carlo* simulations are presented. First, Markov chains are introduced and the fundamental *Master equation* is explained. Next, a simple Markov chain for the Bernoulli process is built. Then, the Metropolis-Hasting algorithm is explained, which allows for sampling according to rather general desired distributions, which is in particular useful for cases where direct sampling does not work. Here again, the Bernoulli process is used as example, for pedagogical reasons, considering that direct sampling is possible there.

In the fourth section, the two main concepts, biasing and Markov chains, are combined. This allows for "blind" sampling, where the process under investigation is driven into the rare-event region of interest, without the need to know how exactly the system has to be "pushed". This is done automatically by using a bias where the bias depends on the quantity of interest. Here, the standard case of an exponential bias, also called exponential tilt, is described. To demonstrate this, the Bernoulli process is considered. Here, unusual quantities of measurements, like the number of consecutive equal coin flips of length of at least 3 are used. This is useful, because the educated sampling does not work for such quantities. Then it is explained, how to obtain the true distribution, i.e., how to remove the bias from the sampled data. This involves also the determination of normalization constants, i.e., partition functions. In the last part, it is explained how the basic algorithm presented so far for the Bernoulli process can be generalized. This yields a "black box" approach, which means that it is applicable for rather arbitrary stochastic processes, at least if the quantity of interest is a scalar variable. Finally, some examples of application of the large-deviation approach to a variety of models are shortly discussed.

## 2 Simple and biased sampling

The most natural implementation of a stochastic system is to sample the states or configuration according to the original distribution of the system. Performing measurements means to take just the configurations, calculate the measurable quantities, and then obtain averages or histograms. This is introduced in the first subsection of this chapter. Next, as an example the Bernoulli process is considered. Then it is shown that one can reach the rare events, i.e., the tails of the distributions, by using biased sampling. Finally, this will be illustrated for the Bernoulli process again.

### 2.1 Simple sampling

In principle, we consider any model which is described by probabilities $P(\underline{y})$ of "objects" $\underline{y}$, e.g, outcome of coin flips, other random numbers, orientation of spins, position of particles, etc.

We want to measure numerically sample averages of measurable quantities $A = A(\underline{y})$. The sample averages are estimates of expectation values

$$\langle A \rangle := \sum_{\underline{y}} A(\underline{y}) P(\underline{y}). \tag{4}$$

As numerical approach, we could use *simple sampling*, i.e., generate $M$ configurations $\underline{y}^{(1)}, \ldots, \underline{y}^{(M)}$ according to the original probabilities $P(.)$. For an experiment, this corresponds to just taking measurements for the given system, as it is. In this case, we have to calculate a sample mean to obtain an estimate for the expectation value

$$\langle A \rangle \approx \frac{1}{M} \sum_i A(\underline{y}^{(i)}).$$

In some cases, one can perform the simple sampling in a *direct* way. This means that one can implement a function or subroutine, which each time it is called, it generates an independent sample of a configuration, which is properly distributed. One example is the *inversion method*, which is based on applying [1] the inverted cumulative distribution function to a random number $r$, which is uniformly distributed in the interval $[0, 1]$. For the latter one, one writes $r \sim U(0, 1)$. If on the other hand no direct sampling approach is available for the considered model, one has to use more involved techniques, in particular Markov chain Monte Carlo simulations, see Chap. 3.

### 2.2 Simple sampling for the Bernoulli process

Let's perform the simple sampling for the Bernoulli process with parameters $\alpha$ and $n$, as defined in Eq. (2). From the algorithmic point of view, we can use direct sampling. This means, we iterate over all entries $y_i$ of $\underline{y}$ and each time we draw a $U(0, 1)$ uniformly distributed number $r$ and if $r < \alpha$ we assign $y_i = 1$ and if $r \geq \alpha$ we assign $y_i = 0$. Then we measure the number $l = l(\underline{y}) = \sum_{i=1}^n y_i$ of 1's. We repeat this many times and build a histogram for the number of observed occurrences of the different values of $l$. Here we expect to obtain approximately the Binomial distribution Eq. (3).

To comply with Eq. (4), the histogram is represented for $k = 0, 1, \ldots, n$ by the measurable quantities

$$A_l(\underline{y}) = \delta_{l,l(\underline{y})} = \delta_{l,\sum_i y_i}. \tag{5}$$

Thus, the sample mean of $A_l$, also called the *empirical measure*, approximates the probability $P_n(l)$ to measure a number $l(\underline{y}) = l$ of 1's in an experiment with $n$ coin flips.

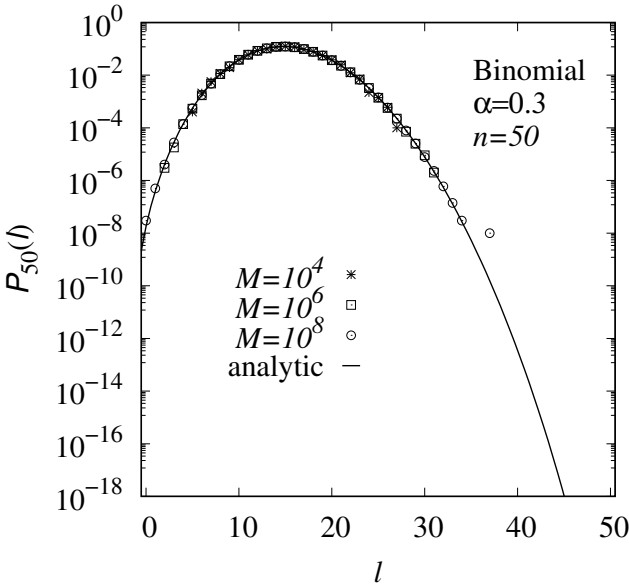

Figure 1: Numerically measured histograms of the number $l$ of 1's obtained from $n = 50$ coin flips. The histograms are measured from $M = 10^4$, $10^6$ and $10^8$ experiments. The solid line shows the analytical solution Eq. (3).

A C code implmentation is provded in the source code [2] `bernoulli_direct.c`.[1]

The result obtained within a simple simulation, here for $n = 50$ and $\alpha = 0.3$, could look like shown in Fig. 1. We consider three different values for the number $M$ of samples, namely $M = 10^4$, $10^6$ and $10^8$. The histograms follow the true distribution Eq. (3) better for increasing $M$, but the tails are not accessible.

To sample also the tails of the distribution, one does not need to raise $M$ to $10^{18}$. Instead, one performs simulations using biased distributions, i.e., not the original ones. This will be explained next.

## 2.3 Biased sampling

To understand the basic principle that allows us to obtain the tails of distributions, we consider another given probability distribution $Q$ for our system of interest. It should have nonzero probabilities $Q(\underline{y}) > 0$ for all configurations $\underline{y}$, but apart from this, it can be arbitrary in principle. This allows us to rewrite Eq. (4) as [3]

$$\langle A \rangle = \sum_{\underline{y}} A(\underline{y}) P(\underline{y}) = \sum_{\underline{y}} A(\underline{y}) \frac{P(\underline{y})}{Q(\underline{y})} Q(\underline{y}) = \langle AP/Q \rangle_Q, \qquad (6)$$

where $\langle \cdots \rangle_Q$ is the average according to $Q$. This means we can generate numerically configurations $\underline{y}^{(i)}$ with simple sampling according to a different distribution $Q$ and will arive in principle at the same result. Since $Q$ is used instead of $P$, one says one has introduced a *bias*, i.e., $Q$ is called the biased distribution. Also, one computes a different sample mean, namely $\{A(\underline{y}^i) P(\underline{y}^i)/Q(\underline{y}^i)\}$ for this biased sampling. Eq. (6) tells us it will give still the same results, for the theoretical case of an analytical evaluation.

Although the biased result aims in principle to estimate the same expectation value $\langle A \rangle$, it may be actually beneficial to use a bias. The basic idea of the biased approach is that by suitable choice of $Q$ one can shift the numerical sampling to a different part of the configuration

---

[1]It can be compiled in a unix shell with `cc -o bernoulli bernoulli_direct.c -lm`.

space. Thus, we could choose that part which is most important for the quantity $A$, e.g., the low-probability tails of the distribution $P$. For this reason, biased sampling is in some fields called *importance sampling*.[2]

As a simple hint why this could be useful, we consider variance reduction. The variance of the distribution of the quantity $A$ of interest is given by

$$\sigma^2(A) := \sum_{\underline{y}} (A(\underline{y}) - \langle A \rangle)^2 P(\underline{y}) = \langle A^2 \rangle - \langle A \rangle^2. \tag{7}$$

This quantity determines, e.g., an estimate of the error of the estimate of $\langle A \rangle$ as $\sigma(A)/\sqrt{(M-1)}$.

Now, when sampling according to $Q$, the relevant variance is for $AP/Q$, which reads as

$$
\begin{aligned}
\sigma_Q^2(AP/Q) &:= \sum_{\underline{y}} \left( \frac{A(\underline{y})P(\underline{y})}{Q(\underline{y})} - \left\langle \frac{AP}{Q} \right\rangle_Q \right)^2 Q(\underline{y}) \\
&= \sum_{\underline{y}} \left( \frac{A(\underline{y})P(\underline{y})}{Q(\underline{y})} - \langle A \rangle \right)^2 Q(\underline{y}).
\end{aligned}
\tag{8}
$$

Now, we want to use this equation to design a suitable distribution $Q$. As extreme artificial case, we assume that $\langle A \rangle$ is known and $A(\underline{y}) \geq 0$ for all possible configurations $\underline{y}$. We choose

$$Q(\underline{y}) = \frac{A(\underline{y})P(\underline{y})}{\langle A \rangle}. \tag{9}$$

When inserting into Eq. (8) one obtains $\sigma_Q^2(AP/Q) = 0$. This means the measurement is arbitrarily accurate! At first sight surprising, this actually holds trivially because in each run of the numerical experiment the value $AP/Q = \langle A \rangle$ is measured, i.e., the desired result. Clearly, this is an rather artificial example, because $\langle A \rangle$ is usually not known. If it was known, one would not have to perform the numerical experiment.

Nevertheless, we can learn a bit from this example. First, Eq. (9) tells us that the biased sampling distribution will usually depend on the quantity of interest. This also means, if we are interested in several quantities for one model, we have to perform several independent simulations with different biases to obtain the full result. If we used only simple, i.e. unbiased sampling, only one set of runs is needed, where jointly all quantities of interest can be obtained.

Second, Eq. (9), since $A$ appears as a multiplicative factor, teaches us that it is favorable to sample more frequently where the measured value is relatively large. This makes sense, because a larger measured value will contribute more to the average. For the case of $A$ measuring a histogram bin as in Eq. (5), it is obvious that we need a considerable amount of data that is located in the desired bin, to estimate the corresponding probability with high accuracy.

## 2.4 Biased sampling for the Bernoulli process

As an example where biased sampling is useful, we aim at numerically estimating over the full support the probabilities $P(l)$ of the number of 1's in the Bernoulli process. Still, the parameter $\alpha$ denotes the the probability of observing a 1 in a single flip. Since we measure $l$ and because the number of 1's is directly correlated to $\alpha$, the basic idea for biasing is to use the original distribution, but for other values $\beta \neq \alpha$ of the parameter. Thus, for the biased sampling probability $Q(\underline{y})$ we use the Bernoulli distribution Eq. (2). Therefore, we do not

---

[2]But in Physics, *importance sampling* is often used to describe the sampling according to the originial, i.e. unbiased, distribution.

have to change the program at all for performing the desired sampling. Only for the analysis according to Eq. (6) we need the ratio $P/Q$ which is

$$P(\underline{y})/Q(\underline{y}) = \frac{\alpha^l(1-\alpha)^{n-l}}{\beta^l(1-\beta)^{n-l}} \,. \tag{10}$$

Note that this ratio only depends on the quantity $l$ of interest, thus it can be directly applied when evaluating the histogram. There is no need to evaluate it for each single sampled configuration $\underline{y}^i$.

The resulting histograms as obtained from simulations for four different values $\beta = 0.1$, 0.3, 0.6, and 0.8, each time for $M = 10^4$ samples, and rescaled according to Eq.(10), is shown in Fig. 2. Note that the different histograms overlap such that the distribution $P(l)$ is estimated over its full support, down to probabilities as small as $10^{-25}$. A very good agreement with the known analytical result is visible. In the tails of the individual rescaled histograms the statistics becomes worse, such that here deviations from the analytic result are visible. If one was interested in obtaining a final single-histogram result, one would therefore clip all raw histograms in the low-statistics tail. For each possible value of $l$, one would use the histogram that exhibits the best statistics.

For this example, obtaining the tails is particularly simple: first, the quantity of interest can be very well controlled by the parameter $\beta$ of the biased sampling distribution $Q$, which here, even better, is the original distribution, just for a different value of the parameter. Since we know how to change the parameters to address to region of interest, one could call this *educated sampling*. Second, both the biasing distribution $Q$ and the original distribution $P$ are known in detail, such that the rescaling by multiplication with $P/Q$ is possible.

For general large-deviation simulations, it is usually hard to relate the measurable quantity $A$ of interest to parameters of the biased sampling distribution. A suitable name for an approach which does not need or have this information could be *blind sampling*. Note that usually the biased sampling distribution is not known completely. In particular, the normalization is unknown, which makes the unbiasing also more difficult, but still possible as we will learn below.

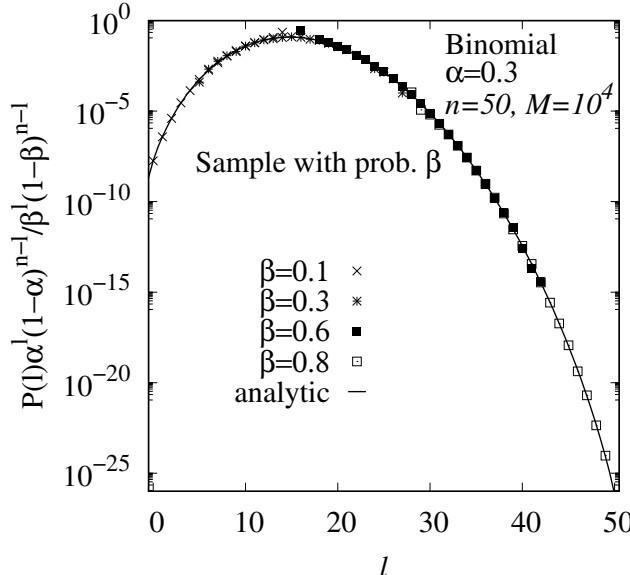

Figure 2: Histograms of the number of 1's obtained from $n = 50$ coin flips for four different values $\beta$ of the probability to obtain a 1, rescaled to result in the distribution for $\alpha = 0.3$. The histograms are obtained from $M = 10^4$ numerical experiments, respectively. The solid line shows the analytic solution Eq. (3).

In order to address these two problems, we first have to be able to sample according to more or less arbitrary distributions with possibly unknown normalization factor. This can be achieved by using *Markov chains*, which are covered next.

## 3   Markov chain Monte Carlo simulations

We consider now a very general setup, a system with a finite number $K$ of states $\underline{y} = \underline{y}_1, \underline{y}_2, \ldots, \underline{y}_K$. The system shall be random. Thus, we assume that the behavior of the system is described by the probability distribution of states represented by probabilities $P(\underline{y}) \geq 0$ with $\sum_{\underline{y}} P(\underline{y}) = 1$.

We still consider the standard target to calculate or estimate the expectation values of observables $A(\underline{y})$

$$\langle A \rangle := \sum_{\underline{y}} A(\underline{y}) P(\underline{y}). \tag{11}$$

Typically, the number $K$ of states is exponentially large as a function of some parameter, e.g. the number $N$ of particles or generally of degrees of freedom. If, for example, each degree $i$ has two states $y_i = 0, 1$ as for the Bernoulli example, we have $K = 2^N$. This means, performing a full enumeration of Eq. (11) is not feasible for even moderate values of $N$.

Often, the most simple numerical estimation procedure is *uniform sampling*. Here one generates a certain number $M$ states $\{\underline{y}^i\}$ ($i = 1, \ldots, M$) randomly, with uniform probability. A typical value of $M$ could be, for example, $10^6$. For the Bernoulli example, one would obtain the uniform sampling by simply choosing for all entries $y_i = 0, 1$ with probability 0.5, instead of the actual probabilities $(1-\alpha)$ and $\alpha$, respectively. To estimate the average $\langle A \rangle$, one calculates the observable for all sampled states and weights the results with the correct probabilities. One has to normalize this weighted average by dividing by the sum of weights. This results in

$$\langle A \rangle \approx \overline{A}^{(1)} := \frac{\sum_{\underline{y}^i} A(\underline{y}^i) P(\underline{y}^i)}{\sum_{\underline{y}^j} P(\underline{y}^j)}. \tag{12}$$

It can be easily checked that this way of normalizing is correct. Say the measurable quantity was boringly constant $A \equiv 1$, then Eq. (12) would result in $\sum_{\underline{y}^i} P(\underline{y}^i) / \left( \sum_{\underline{y}^j} P(\underline{y}^j) \right) = 1$ which is correct. Also, if the sample consisted of all $K$ possible states, then the sum $\sum_{\underline{y}^i} P(\underline{y}^i)$ in the denominator is one by normalization and Eq. (12) reduces to the exact expectation value Eq. (11). Note also that for Eq. (12) it is sufficient to know the probabilities without normalization, since this cancels from the ratio.

Unfortunately, the behavior of systems of interest is typically concentrated around some exponentially small fraction of states. This means that $P(\underline{y})$ is significantly large only for this fraction. If, as an arbitrary example, the number of relevant states grows exponentially like $2^{N/2}$, but the number of states like $2^N$, the fraction of relevant states would decrease as $2^{N/2}/2^N = 2^{-N/2}$. A physical example is an Ising ferromagnet at low temperature. Here the number of states is indeed $2^N$ but at low temperatures most states have a finite magnetization, i.e., most spins have the same sign. Here the fraction of relevant states is even smaller than $2^{-N/2}$. Physically speaking, a uniformly sampled random state would basically never exhibit a significant magnetization.

Thus, with uniform sampling one would almost sure obtain states $\underline{y}^i$ where $P(\underline{y}^i)$ is exponentially smaller than the probability of relevant states, which means one misses the main contributions to the estimation of the average and $\overline{A}^{(1)}$ is not very accurate.

A much better approach is to generate $M$ configurations $\underline{y}^i$ according to the desired probabilities $P(\underline{y}^i)$, i.e., direct sampling, as already introduced in Sec. 2.1. In this case, the sampling ensures correct statistics and the estimation of the expectation value $\langle A \rangle$ reduces to the empirical average

$$\langle A \rangle \approx \overline{A}^{(2)} := \sum_{\underline{y}^i} A(\underline{y}^i)/M \qquad (\text{with } \underline{y}^i \sim P(\underline{y}^i)). \tag{13}$$

Note that $P$ denoted so far the original probabilities of the system. But, in the present context, it can be the biased probabilities $Q$ to drive the system to desired but unlikely configurations, as done for large-deviation sampling in section 2.3. In this case, one would have to chose $A$ appropriately to unbias the result, as discussed previously. We will come back to large-deviation sampling later in section 4, but for the moment we assume just arbitrary given probabilities $P$ according which we desire to sample, let them be the original or biased ones.

A direct sampling can be achieved for very simple models, often by the inversion method. This means, one can write a function in the computer code which upon each call returns a statistically independent configuration according to the desired statistics. This is indeed possible for the Bernoulli process, where each coin flip can be simply chosen randomly. Unfortunately, for most systems of interest, in particular if one applies biases which depend on complex measurable quantities, no general approach is known, where one can perform this direct sampling as to obtain one independently drawn configuration from each function call.

A general solution for this sampling problem is provided by the Markov chain Monte Carlo approach, which is introduced in the next section. As we will see, it will indeed allow for sampling according to arbitrary distributions, at least in principle. But this comes at the price that correlations between the sampled configurations are generated, such that only some, possible very small, fraction of the sampled states are statistically independent.

## 3.1 Markov chains

When defining *Markov chain* one considers a set $\mathcal{S}$ of states of some elements. The set can be finite, countable, or uncountable. For our applications, i.e. in the following, we are interested in states that are configurations $\underline{y} \in \mathcal{S}$ of a system described by the values of several variables $\underline{y} = (y_1, \ldots, y_n)$, e.g. the result of $n$ coin flips.

Now, a Markov chain is a sequence $\underline{y}(0) \rightarrow \underline{y}(1) \rightarrow \underline{y}(2) \rightarrow \ldots$ of states $\underline{y}(t)$ at (here) discrete times $t = 0, 1, 2, \ldots$ The sequence is generated by a *probabilistic dynamic*, as explained now. The main property of Markov chains is that state $\underline{y}(t+1)$ depends in a stochastic way only on its preceding state $\underline{y}(t)$, but not on states earlier in the chain.

In case of a countable number of possible states, one formally describes the transitions $\underline{y}(t) \rightarrow \underline{y}(t+1)$ by a *transition matrix* $W_{\underline{y}\underline{z}} \equiv W(\underline{y} \rightarrow \underline{z})$ which states the probability[3] to move from state $\underline{y}$ (at time $t$) to state $\underline{z}$ (at time $t+1$). Usually, one considers the case that $W_{\underline{y}\underline{z}}$ does not depend on time.

Being probabilities, the following simple properties of $W_{\underline{y}\underline{z}}$ arise:

$$\begin{aligned} W_{\underline{y}\underline{z}} \geq 0\,, & \qquad \forall \underline{y}, \underline{z} \in \mathcal{S} & (\textit{positivity})\,, \\ \sum_{\underline{z}} W_{\underline{y}\underline{z}} = 1\,, & \qquad \forall \underline{y} \in \mathcal{S} & (\textit{conservation})\,. \end{aligned} \tag{14}$$

The combination of state space and the transition probabilities is called a *Markov process*.

---

[3]One can also define Markov chains in continuous time. Then one would use transition rates instead of transition probabilities.

When simulating a Markov process on a computer, one speaks of a *Markov chain Monte Carlo* (MCMC) simulation.

To understand better what happens when a Markov chain is created and to investigate the dynamics of generating, or "running" a Markov chain, we assume that one does not run not only one but a certain number $N_{\text{tot}} \gg 1$ of independent Markov chains. Then one records $N(\underline{y}, t)$, which is here the number of chains that are in state $\underline{y}$, at time step $t$.

---

Example: Two state system

The most simple system consists of two states, here called A and B with four transition probabilities. Here we assume some arbitrary values $W_{AA} = 0.6$, $W_{AB} = 0.4$, $W_{BA} = 0.1$, $W_{BB} = 0.9$. Thus, the probability to stay in state A is 0.6, while the probability to leave from A to B is 0.4.

Here we assume that we run $N_{\text{tot}} = 100$ Markov chains that all start in state *A*. Therefore $N(A, 0) = 100$ and $N(B, 0) = 0$. How could the dynamics look like for these 100 chains?

In the first transition step, about $N(A, 0)W_{AB} = 100 \times 0.4 = 40$ chains may move from $A \to B$, while the other chains remain in A and no transition $B \to A$ happens at $t = 0$, since no chain is in state B. This can be depicted as:

$$t = 0: \quad \boxed{\text{N(A)=100}} \overset{0}{\underset{40}{\rightleftarrows}} \boxed{\text{N(B)=0}}.$$

Note that the number of transitions $A \to B$ is a random quantity, so it may deviate substantially from 40 for 100 chains. But for the purpose of simplicity, we work with typical values here.

Now, at $t = 1$ we have $N(A, 1) = 60$ and $N(B, 1) = 40$. Therefore, $N(A, 1)W_{AB} = 60 \times 0.4 = 24$ chains may move from $A \to B$, while $N(B, 1)W_{BA} = 40 \times 0.1 = 4$ may move from $B \to A$, while the other chains exhibit no change of state:

$$t = 1: \quad \boxed{\text{N(A)=60}} \overset{4}{\underset{24}{\rightleftarrows}} \boxed{\text{N(B)=40}}.$$

Now, at $t = 2$ we have $N(A, 2) = 40$ and $N(B, 2) = 60$. The subsequent dynamics until $t = 7$, again when considering only the most likely evolution, may look as follows:

$$t = 2: \quad \boxed{\text{N(A)=40}} \overset{6}{\underset{16}{\rightleftarrows}} \boxed{\text{N(B)=60}},$$

$$t = 3: \quad \boxed{\text{N(A)=30}} \overset{7}{\underset{12}{\rightleftarrows}} \boxed{\text{N(B)=70}},$$

$$t = 4: \quad \boxed{\text{N(A)=25}} \overset{8}{\underset{10}{\rightleftarrows}} \boxed{\text{N(B)=75}},$$

$$t = 5: \quad \boxed{\text{N(A)=23}} \overset{8}{\underset{9}{\rightleftarrows}} \boxed{\text{N(B)=77}},$$

$$t = 6: \quad \boxed{\text{N(A)=22}} \quad \overset{8}{\underset{9}{\overset{\leftarrow}{\longrightarrow}}} \quad \boxed{\text{N(B)=78}},$$

$$t = 7: \quad \boxed{\text{N(A)=21}} \quad \overset{8}{\underset{8}{\overset{\leftarrow}{\longrightarrow}}} \quad \boxed{\text{N(B)=79}}.$$

From now on one has about $N(A,t)W_{AB} = N(B,t)W_{BA}$. This means, the transfer between the two states balances leading to $N(Y,t) =$const for $Y = A$ and $Y = B$. Thus, a *stationary state* is reached and $N(Y,t) = N(Y)$ will become time-independent when ignoring relatively small fluctuations, at least if the number $N_{\text{tot}}$ of states is very large. □

To allow for a formal description of the dynamics of a Markov chain, we define $P(\underline{y},t) = \lim_{N_{\text{tot}}\to\infty} \langle N(\underline{y},t)/N_{\text{tot}} \rangle$ be the probability that the system at time $t$ is in state $\underline{y}(\overline{t}) = \underline{y}$.

We want to write down a balance equation for the probability $P(\underline{y},t)$ to be in state $\underline{y}$. Usually, this fundamental equation is called *Master Equation*. State $\underline{y}$ will have connections, i.e. non-zero transition probabilities to other states, see Fig. 3. This may lead to some chains move into state $\underline{y}$, i.e. a flow of probability into state $\underline{y}$, which results in a contribution increasing $P(\underline{y},t+1)$ with respect to $P(\underline{y},t)$. This flow of probability will be proportional to the probability being in another state $\underline{z}$ and proportional to the transition probability $W_{\underline{z}\underline{y}}$, thus proportional to $W_{\underline{z}\underline{y}}P(\underline{z},t)$. On the other hand, some chains will be in state $\underline{y}$ and move out to other states $\underline{z}$. Hence, this flow of probability is given by $W_{\underline{y}\underline{z}}P(\underline{y},t)$ and it contributes negatively to $P(\underline{y},t+1)$.

This leads to the following equation for the change $\Delta P(\underline{y},t)$ of the probability from time $t$ to $t+1$:

$$\Delta P(\underline{y},t) := P(\underline{y},t+1) - P(\underline{y},t) = \sum_{\underline{z}} W_{\underline{z}\underline{y}}P(\underline{z},t) - \sum_{\underline{z}} W_{\underline{y}\underline{z}}P(\underline{y},t), \qquad \forall \underline{y} \in \mathcal{S}. \quad (15)$$

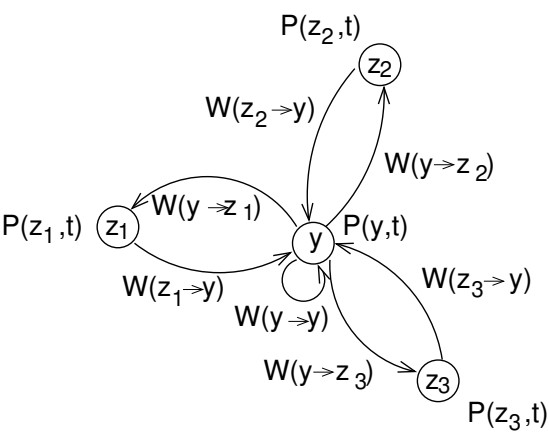

Figure 3: Flow of probability from and to state $y$. This and other states are shown as circles. It is assumed that $y$ has non-zero transition probabilities to and from three other states $z_1$, $z_2$ and $z_3$, shown as arrows. Other possible transitions of the states $z_1$, $z_2$ and $z_3$ are not shown here.

Under specific conditions [4], as specified by the Theorems of Perron [5] and Frobenius [6], the probability distribution will converge to a stationary, i.e. time-independent distribution

$$P_{\text{st}}(\underline{y}) := \lim_{t \to \infty} P(\underline{y}, t).\tag{16}$$

This will happen in particular if largest left eigenvalue $\lambda = 1$ of the matrix $W$ has multiplicity 1, i.e., the corresponding eigenspace is one-dimensional.

The limit Eq. (16) has to be independent of any given initial state $\underline{y}(0)$. The Markov process is called *ergodic* in this case. This basically means that one can reach each state from any other state by a finite sequence of transitions. Note that there are non-ergodic systems, a simple example is shown in Fig. 4.

Now, after having introduced Markov chains in a general way, we come back to the target of sampling configurations $\underline{y}$ of a system according to given probabilities $P(\underline{y})$. For a Markov chain that achieves a stationary probability distribution $P_{\text{st}}$ this can be achieved by

$$\boxed{\text{choose } W_{\underline{y}\underline{z}} \text{ such that } P_{\text{st}} = P.}$$

Since $P_{\text{st}} = P$ is time-independent we simply replace $P(\underline{y}, t)$ by $P(\underline{y})$. This implies from Eq. (15) that

$$0 = \Delta P(\underline{y}) = \sum_{\underline{z}} W_{\underline{z}\underline{y}} P(\underline{z}) - \sum_{\underline{z}} W_{\underline{y}\underline{z}} P(\underline{y}), \qquad \forall \underline{y} \in \mathcal{S}.$$

This means one has obtained more conditions for the transition probabilities, in addition to Eqs. (14), depending on the target probabilities $P(\underline{y})$. These conditions are called *global balance*, because for each state the total inflow of probability, globally summed over all other states, balances with the total outflow. In particular, the global balance conditions relate the transition probabilities to the desired sampling probabilities $P(\underline{y})$.

A very convenient way to fulfill the global balance is to make each pair of corresponding contributions of the two sums cancelling each other, i.e.,

$$W_{\underline{z}\underline{y}} P(\underline{z}) - W_{\underline{y}\underline{z}} P(\underline{y}) = 0, \qquad \forall \underline{y}, \underline{z} \in \mathcal{S}.\tag{17}$$

This is called *detailed balance*, because the cancelation of probability flow appears for each pair of states. These conditions help a great deal in setting up suitable transition probabilities that lead to Markov chains generating configurations according to the target probabilities $P(\underline{y})$. One special widely-used algorithm, the Metropolis-Hastings algorithm, will be presented in Sec. 3.3.

In general and even if one takes the Metropolis-Hastings algorithm, there is a lot of freedom in choosing the transition probabilities, influencing strongly the efficiency of the corresponding algorithm. One general rule is, the more the configurations are changed between two time steps, the faster the algorithm will walk through configuration space, thus the more efficient

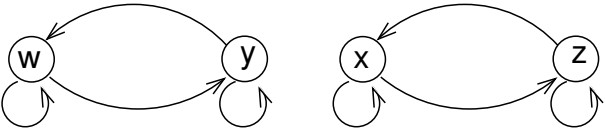

Figure 4: Example for a non ergodic system. Possible transitions with non-zero transition probabilities are indicated by arrows. A Markov chain which starts in $y$=W or $y$ = Y will always have $P(\text{X}, t) = 0$ and $P(\text{Z}, t) = 0$ while this will not be the case if the chain started in X or Z. Thus, the limiting distribution will depend on the initial state.

the algorithm is. Unfortunately, these system-wide changes are hard to set up, often one has to be satisfied with single-variable changes.

The efficiency of the Markov chain Monte Carlo approach is visible through two properties: the speed of *equilibration* and through the occuring temporal *correlations*. Equilibration refers to the fact that the Markov chains start in some configuration $y(0)$. Since typically, the configurations are changed only a bit between two consecutive steps of the Markov chain, it will take a certain number of steps $t_{\text{equi}}$ to "forget" this initial configuration. Thus, when calculating averages, the configurations obtained for $t < t_{\text{equi}}$ have to be omitted. In the next section, an example is shown which illustrates how the equilibration becomes visible in a simulation outcome, allowing one to estimate $t_{\text{equi}}$.

The fact that configurations change only gradually over time, means $y(t+1)$ is typically similar to $y(t)$. Thus, configurations which are close in time $t$ are correlated even after equilibration. Since one is interested in sampling statistically independent configurations, only distant states $y(t), y(t+\Delta t), y(t+2\Delta t), \ldots$ should be included in the measurements.

Note that $t_{\text{equi}}$ and $\Delta t$ depend strongly on the considered system, on the applied algorithm and on the parameters used for the simulation. Usually, they cannot be calculated in advance but have to be determined experimentally in exploratory test simulations. Some systems, in particular if they are large, are too hard to simulate such that even equilibration cannot be achieved in reasonable time with the algorithms at hand.

## 3.2 MCMC for the Bernoulli process

As a simple toy application of Markov chain Monte Carlo simulations, we consider again the Bernoulli process, i.e., configurations $y = (y_1, y_2, \ldots, y_n)$ of $n$ coin tosses $y_i \in \{0, 1\}$. Like before, we use the parameter $\alpha \in [0, 1]$ to define the probabilities for the two possible outcomes of each coin toss, see Eq. (1).

Now, we want to use a Markov chain to sample states according to the configuration probabilities Eq. (2). Note that for this simple model, a Markov chain is actually not needed, since one can simulate the coin tosses efficiently by directly sampling, as done in Sec. 2.2. Thus, this simple application is for pedagogical reasons only.

As mentioned before, there are in general many possibilities to set up the transition probabilities in the matrix $W$. Here we proceed as follows: Given the current state $y = y(t)$, we just select $n_{\text{c}}$ times a randomly chosen entry and redraw it according to the original probabilities. Therefore, we perform $n_{\text{c}}$ new coin tosses, while the outcomes of the other coin tosses contained in $y$ are not touched. This is implemented by the following algorithm:

**algorithm** Bernoulli-MC($y$, $n$, $n_{\text{c}}$)
**begin**
   **do** $n_{\text{c}}$ times:
   **begin**
      choose random index $i \in \{1, \ldots, n\}$
      redraw $y_i$ according to (1)
   **end**
**end**

The resulting configuration is $y(t+1)$. Each time a new index $i$ is drawn randomly and independently, this might result in repeated redrawing some entries. Since each coin toss is independent, only the last one will be effective for a given entry. Thus, the actual number of entries actually redrawn may be smaller than $n_{\text{c}}$. An implementation in C is contained [2] in `mc_bernoulli.c`[4] in the function `bernoulli_mc_step0()`.

---

[4]The code can be compiled in a Unix shell by `cc -o mc_bernoulli mc_bernoulli.c -lm`.

It is intuitively clear that changing some entries according to the original probabilities, while leaving all others untouched, which nevertheless have previously been drawn in this way, will lead to the correct sampling. Still, we want to check detailed balance explicitly. The desired sampling probability of configuration $\underline{y}$ is given by Eq. (2).

For obtaining the transition probabilities, we consider two states $\underline{y}$ and $\underline{z}$ that are connected by a step in the Markov chain. Let $I_c$ contain those indices where $\underline{y}$ and $\underline{z}$ differ, i.e., $I_c = \{i | y_i \neq z_i; i = 1, \ldots, n\}$. Note that $\underline{y}$ and $\underline{z}$ can have a non-zero transition probability only when $n_c \geq |I_c|$.

Furthermore, let $I_s$ be the actual set of indices selected by the algorithm, with $|I_s| \leq n_c$. This set will most of the time be a super set of $I_c$, because for some of the selected entries redoing the coin flip might lead to the previous outcome. Hence, for the entries in $I_s \setminus I_c$, the entries are not changed, like those entries which do not appear in $I_s$ anyway. We denote by $R(I_s)$ the probability that the algorithm selects a certain subset $I_s$, which might be a complicated function of $I_s$. For the entries $i$ which appear in $I_s$, the joint probability for obtaining the outcomes $z_i$ is simply $\prod_{i \in I_s} \alpha^{z_i}(1-\alpha)^{1-z_i}$. Splitting $I_s$ into $I_c$ and $I_s \setminus I_c$, and summing over all possible sets $I_s$, this leads to the following transition probability:

$$W_{\underline{y}\underline{z}} = \sum_{I_s} R(I_s) \prod_{i \in I_s \setminus I_c} \alpha^{z_i}(1-\alpha)^{1-z_i} \prod_{i \in I_c} \alpha^{z_i}(1-\alpha)^{1-z_i} .$$

Since $I_c$ is fixed, the last product factor is the same for all terms in the sum. Thus, it can be taken out of the sum, leading to

$$W_{\underline{y}\underline{z}} = \prod_{i \in I_c} \alpha^{z_i}(1-\alpha)^{1-z_i} \left( \sum_{I_s} R(I_s) \prod_{i \in I_s \setminus I_c} \alpha^{z_i}(1-\alpha)^{1-z_i} \right) .$$

The transition probability for the opposite move is obtained by replacing $z_i$ with $y_i$. The selected subsets $I_s$ which contribute to the transition are the same as for the forward move. This means they have the same probabilities $R(I_s)$. Therefore, as we will see below, that we do not have to know the function dependency of $R(I_s)$:

Let's now check detailed balance Eq. (17) explicitly. For this purpose, we split the product of $P(\underline{y})$ into two factors, containing contributions from $I_c$ and from all other entries. Also, we use that $y_i = z_i$ for $i \notin I_c$, in particular for $i \in I_s \setminus I_c$:

$$P(\underline{y})W_{\underline{y}\underline{z}} = \underbrace{\prod_{i \in I_c} \alpha^{y_i}(1-\alpha)^{1-y_i}}_{A} \underbrace{\prod_{i \neq I_c} \alpha^{y_i}(1-\alpha)^{1-y_i}}_{B}$$

$$\times \underbrace{\prod_{i \in I_c} \alpha^{z_i}(1-\alpha)^{1-z_i}}_{C} \underbrace{\left( \sum_{I_s} R(I_s) \prod_{i \in I_s \setminus I_c} \alpha^{z_i}(1-\alpha)^{1-z_i} \right)}_{D}$$

$$\overset{y_i = z_i; i \neq I_c}{=} \prod_{i \in I_c} \alpha^{z_i}(1-\alpha)^{1-z_i} \prod_{i \neq I_c} \alpha^{z_i}(1-\alpha)^{1-z_i}$$

$$\times \prod_{i \in I_c} \alpha^{y_i}(1-\alpha)^{1-y_i} \left( \sum_{I_s} R(I_s) \prod_{i \in I_s \setminus I_c} \alpha^{y_i}(1-\alpha)^{1-y_i} \right)$$

$$\overset{(*)}{=} P(\underline{z})W_{\underline{z}\underline{y}} ,$$

where from the second to the third expression, the factors $A$ and $C$ were just exchanged, and for factors $B$ and $D$ $y_i = z_i$ for $i \notin I_c$ was used. Thus, detailed balance holds.

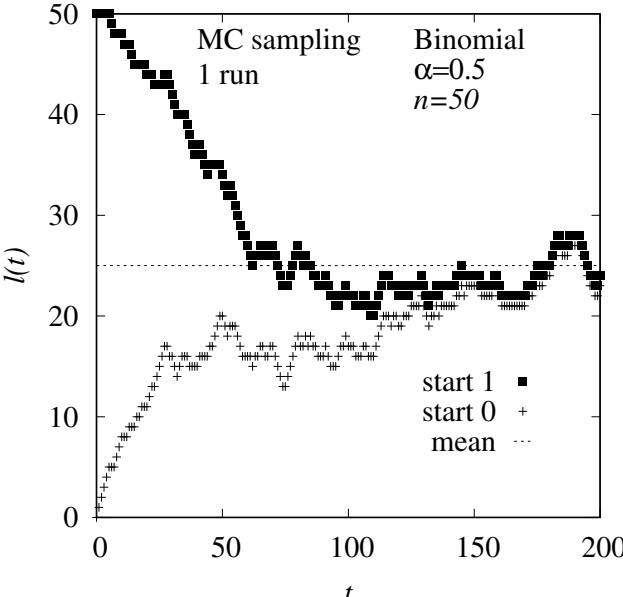

Figure 5: Sample Markov chain for the Bernoulli case for $n = 50$ coin flips: Number $l(t) = \sum_i y_i(t)$ of 1's as function of the Monte Carlo step $t$ for probability $\alpha = 0.5$ and $n_{\mathrm{c}} = 2$. Two extreme different initial configurations are taken into account. The horizontal line indicates the expectation value $\alpha n$.

Now the equilibration of the Monte Carlo simulation is considered. To investigate this, two extreme start configuration $\underline{y}(0) = (y^0, y^0, \ldots, y^0)$ are used, with all entries being the same, either all $y^0 = 0$ or all $y^0 = 1$. Fig. 5 shows the time evolution of the number $l$ of 1's as function of the Monte Carlo step $t$ for the case of $n_{\mathrm{c}} = 2$ changes per Monte Carlo step. After about $t = 150$ steps, the two simulations, which were generated with the same set of random numbers, start fluctuating both around the expectation value $\alpha n = 25$. Thus, the equilibration time $t_{\mathrm{equi}}$ is about 150 steps.

In Fig. 6 $l(t)$ is shown for different number $n_{\mathrm{c}}$ of selected entries for changes. The more entries are changed, the faster the convergence is. This is in general true for all MCMC algorithms: the approach is the more efficient, the faster it can move through configuration space.

## 3.3 Metropolis-Hastings algorithm

Often it is not possible to set up the transition probabilities $W_{\underline{y}\underline{z}}$ directly. This is typically the case for more realistic models or more complex distributions. In particular, when we use large-deviation sampling where the bias depends on the measured quantity of interest, this will occur, as explained in Sec. 4.

For such cases, the *Metropolis-Hastings* algorithm [7, 8] provides a general framework, which makes it way simpler to obtain correct transition probabilities for general problems. The algorithm works well even for the case, where the normalization constant of the target probabilities $P(\underline{y})$ is unknown.

The basic idea works as follow. We assume that we are given the current configuration $\underline{y} = \underline{y}(t)$ of the Markov chain. A Monte Carlo step consists of two parts, which are defined by *two* matrices $A(\underline{y} \to \underline{z})$ and $\tilde{W}(\underline{y} \to \underline{z})$ of probabilities:

1. Select a *trial configuration $\underline{z}$ randomly*, defined by an "arbitrarily" chosen matrix $A(\underline{y} \to \underline{z})$.

2. With probability $\tilde{W}(\underline{y} \to \underline{z})$, the configuration $\underline{z}$ is *accepted*, i.e., $\underline{y}(t+1) = \underline{z}$. With probability $1 - \tilde{W}(\underline{y} \to \underline{z})$, configuration $\underline{z}$ is *rejected*, i.e., $\underline{y}(t+1) = \underline{y}$.

Note that the matrix $A(\underline{y} \to \underline{z})$ is not to be confused with the measurable quantity $A(\underline{y})$. $\tilde{W}(\underline{y} \to \underline{z})$ is called *acceptance probability*. At first sight, it might seem that one has not gained anything, because instead of selecting the transition probabilities $W_{\underline{y}\underline{z}} = W(\underline{y} \to \underline{z})$ one has to choose two probability matrices $A(\underline{y} \to \underline{z})$ and $\tilde{W}(\underline{y} \to \underline{z})$. The main advantage of this approach is, as we will see soon, that one is basically free to choose $A$ and then $\tilde{W}$ is set up such that the correct sampling according to the desired target probabilities $P(\underline{y})$ is ensured.

To understand why this works, we first realize that the two-step process makes the total transition probability a product of the probabilities of the two steps.

$$W(\underline{y} \to \underline{z}) = A(\underline{y} \to \underline{z})\tilde{W}(\underline{y} \to \underline{z}) \qquad (\underline{y} \neq \underline{z}). \tag{18}$$

This means, the total probability to stay in $\underline{y}$ is given by $W(\underline{y} \to \underline{y}) = 1 - \sum_{\underline{z} \neq \underline{y}} W(\underline{y} \to \underline{z})$.

Now, we insert Eq. (18) into the detailed balance condition Eq. (17). We assume that $A$ is somehow given and we resolve with respect to the unknown quantities $\tilde{W}$ and obtain

$$\frac{\tilde{W}(\underline{y} \to \underline{z})}{\tilde{W}(\underline{z} \to \underline{y})} = \frac{P(\underline{z})}{P(\underline{y})} \frac{A(\underline{z} \to \underline{y})}{A(\underline{y} \to \underline{z})}. \tag{19}$$

Now we have to select some acceptance probability matrix $\tilde{W}$ that fulfills this equation. For the Metropolis-Hastings algorithm [7,8] the choice is

$$\tilde{W}(\underline{y} \to \underline{z}) = \min\left(1, \underbrace{\frac{P(\underline{z})}{P(\underline{y})} \frac{A(\underline{z} \to \underline{y})}{A(\underline{y} \to \underline{z})}}_{=:Q}\right). \tag{20}$$

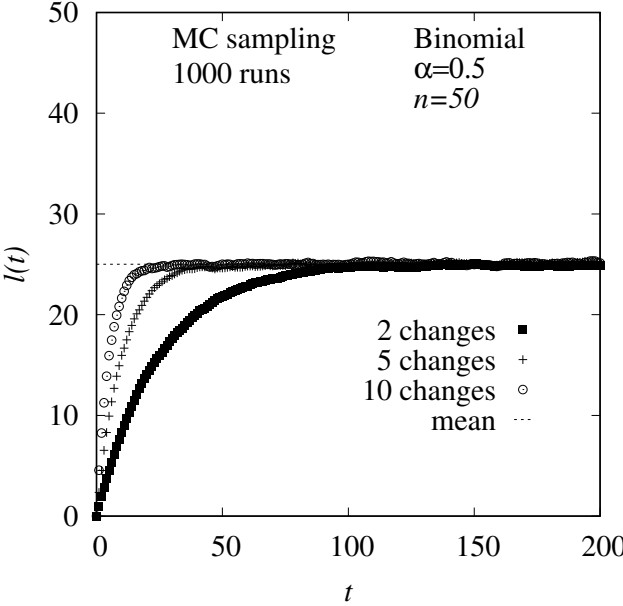

Figure 6: Markov chain for the Bernoulli case with $n = 50$ coin flips and $\alpha = 0.5$. Shown is the number $l(t) = \sum_i y_i(t)$ of 1's, averaged over 1000 runs, starting at $l(0) = 0$. Three different numbers $n_c = 2, 5, 10$ of entry changes are considered. The horizontal line indicates the expectation value $\alpha n$.

Note that for the opposite move $\underline{z} \to \underline{y}$ the vectors $\underline{z}$ and $\underline{y}$ exchange and therefore $Q$ is replaced by $1/Q$. This also means that for one of the two moves between $\underline{z}$ and $\underline{y}$ the transition probability is one. One could use smaller acceptance probabilities, e.g. $1/2$ instead of 1 for one direction, the first term of the min{...}. But then one would have, to conserve detailed balance, to use the half of the probabilities for the opposite direction as well resulting in $\tilde{W}(\underline{y} \to \underline{z}) = \min\left(\frac{1}{2}, \frac{1}{2}\frac{P(\underline{z})}{P(\underline{y})}\frac{A(\underline{z}\to\underline{y})}{A(\underline{y}\to\underline{z})}\right)$, but this would reduce the overall number of performed changes. On the other hand, it is not possible to increase the acceptance rate 1 for the first term in the min{...}. In this sense, the Metropolis-Hastings choice is maximal with respect to the number of accepted moves, given the trial construction probabilities $A(\underline{z} \to \underline{y})$.

We can now verify that the rewritten detailed balance condition holds. Without loss of generality we assume for the ratio $Q < 1$. Then we have

$$
\begin{aligned}
\tilde{W}(\underline{y} \to \underline{z}) &= Q \\
\tilde{W}(\underline{z} \to \underline{y}) &= 1
\end{aligned}
\qquad \Rightarrow \qquad
\frac{\tilde{W}(\underline{y} \to \underline{z})}{\tilde{W}(\underline{z} \to \underline{y})} = Q/1 = Q \,,
$$

as required by Eq. (19). The space of possible algorithms is represented by the trial construction probabilities $A(\underline{y} \to \underline{z})$, thus it is very large. For sure, these values have to be selected such that the resulting Markov process is ergodic. Thus, the probabilities have to be set up in a way such that between any pair of states $\underline{y}$ and $\underline{z}$ there is either a direct transition possible with $A(\underline{y} \to \underline{z}) > 0$, or there exists a finite sequence $x_1, \ldots, x_k$ of intermediate states such that $A(\underline{y} \to \underline{x}_1) > 0$, $A(\underline{x}_i \to \underline{x}_{i+1}) > 0$ for $i = 1, \ldots, k-1$ and $A(\underline{x}_k \to \underline{z}) > 0$.

Typically, the generation of the trial configuration is implemented by changing one or several randomly chosen entries with respect to the current configuration $\underline{y}(t)$. In general, the more entries are changed, while keeping the acceptance probability $\tilde{W}$ high, the faster the algorithm. Unfortunately, there is usually a trade off between acceptance probability and how much the trial state configuration differs from the current configuration. The most simple and often used approach is to change just one entry of the current configuration to obtain the trial one. This is done in the example in the following section.

## 3.4 Metropolis-Hastings algorithm for the Bernoulli process

Again, we consider as simplest toy example the Bernoulli process for $n$ coin tosses $\underline{y} \in \{0,1\}^n$ with probability $\alpha$ for obtaining a 1, see Eq. (1), described by $P(\underline{y})$ according to Eq. (2).

Let $\underline{y} = \underline{y}(t)$ be the current configuration of a Markov chain at step $t$. A very simple *choice* for the generation of a trial configuration $\underline{z}$ is:

- Choose one entry $i_0 \in \{1, \ldots, n\}$ with uniform probability $1/n$.

-
$$
\text{Let} \quad z_i = \begin{cases} 1 - y_i, & i = i_0\,, \\ y_i, & i \neq i_0\,. \end{cases}
\tag{21}
$$

Since exactly one entry is changed with respect to $\underline{z}$, this is called a *single variable flip* algorithm. In statistical physics, e.g., for Ising spin systems like ferromagnets or spin glasses, it is called the *single spin flip* algorithm.

The corresponding trial construction matrix is $A(\underline{y} \to \underline{z}) = A(\underline{z} \to \underline{y}) = 1/n$, if $\underline{y}$ and $\underline{z}$ differ by exactly one entry. For all other cases we have $A(\underline{y} \to \underline{z}) = A(\underline{z} \to \underline{y}) = 0$. For the latter case, no transitions occur between the pairs of configurations, so detailed balance holds trivially. We only have to consider the case where $\underline{y}$ and $\underline{z}$ differ by one entry.

The calculation of the Metropolis acceptance probability Eq. (20), using Eq. (21) yields

$$
\begin{aligned}
\tilde{W}(\underline{y} \to \underline{z}) &= \min\left(1, \frac{\prod_{i=1}^{n} \alpha^{z_i}(1-\alpha)^{1-z_i}}{\prod_{i=1}^{n} \alpha^{y_i}(1-\alpha)^{1-y_i}} \frac{1/n}{1/n}\right) \\
&= \min\left(1, \frac{\alpha^{z_{i_0}}(1-\alpha)^{1-z_{i_0}}}{\alpha^{y_{i_0}}(1-\alpha)^{1-y_{i_0}}}\right) \\
&= \min\left(1, \frac{\alpha^{1-y_{i_0}}(1-\alpha)^{y_{i_0}}}{\alpha^{y_{i_0}}(1-\alpha)^{1-y_{i_0}}}\right) \\
&= \min\left(1, \alpha^{1-2y_{i_0}}(1-\alpha)^{2y_{i_0}-1}\right).
\end{aligned}
$$

Hence, one can calculate the acceptance probability without performing the actual flip beforehand. Thus, one performs a single-variable flip with probability $\tilde{W}$ to yield $\underline{y}(t+1)$, otherwise the current configuration $\underline{y}(t)$ is kept. A corresponding C code [2] is contained in `mc_bernoulli.c` in the function `bernoulli_metropolis()`.

The advantage of changing only a single variable is that such an approach works for most systems, in particular complex systems, where no direct sampling is possible. Still, since at most one variable is flipped in each Monte Carlo step, the algorithm is rather slow. In principle it can be made faster if more than one variable is changed within on step, yielding faster equilibration and a smaller correlation time. But if too many changes are performed, the acceptance rate becomes lower.

## 4 Biased Markov chain sampling

Now we come back to the main target, to obtain for a random process the distribution $Q(S)$ of a measurable scalar quantity $S$, which we also call *score*. The score can be in principle any scalar function of the entire configuration $\underline{y}$. The distribution can be obtained in principle by considering all possible configurations $\underline{y}$ and adding up the probabilities $P(\underline{y})$ of those configurations for which $S(\underline{y})$ has the desired value $S$, i.e.

$$
Q(S) = \sum_{\underline{y}} \delta_{S,S(\underline{y})} P(\underline{y}). \tag{22}
$$

We aim at obtaining $Q(S)$ over a large range of the support, down to the tails, where the probabilities are very small like $10^{-50}$ or even lower.

Again, like in Sec. 2.3, we introduce a *bias* that changes the sampling distribution such that it is shifted to the region of interest. For the previous approach, when applied to the Bernoulli process, we knew that by using a different coin-flip probability $\beta \neq \alpha$ we could shift the distribution $P(l)$ of the number $l$ of 1's to the tails. We called this *educated* sampling, since we know how we have to change the natural parameter of the distribution to concentrate the sampling in the region of interest. In contrast to this, we will now consider the more general case, where we have no idea of how we could influence the sampling by just changing some parameters of the system. Often, as we will show in an example below, there is even no suitable connection of $S$ to system parameters. This more general approach, which allows one to sample the region of interest without prior knowledge, we call *blind sampling*.

The basic idea of this blind sampling is to make the bias depend on the quantity $S$ of interest and some control parameter. This will drive the biased simulation automatically in the desired region, but we do not have to know how the system has to be controlled to do this.

## 4.1 Exponential bias

A standard approach, and for some applications even the optimal one, is to use an *exponential bias*, which for a configuration $\underline{y}$ reads

$$B_\Theta(\underline{y}) = \exp(-S(\underline{y})/\Theta), \tag{23}$$

where $S(\underline{y})$ is the score evaluated for the configuration. This bias appears very natural for physicists since it corresponds to the Boltzmann weight of the *canonical ensemble*. Therefore $\Theta$ is a temperature-like parameter, short just temperature, which controls the impact of the bias. Note that also other biases are possible, in principle everything that works is good. Nevertheless, for many systems the exponential bias works very well and it is also used for analytical calculations, where one can take advantage of biases to obtain tails of distributions in a similar way. For a short introduction, see Ref. [9]. Also more detailed texts, still well readable for physicists, are available [10, 11].

We consider the case that the full distribution of configurations is obtained by the original probability $P(\underline{y})$ times the bias. Like any distribution, it should be normalized, leading to the probability $P_\Theta(\underline{y})$ to obtain a configuration $\underline{y}$ in the biased ensemble as given by

$$P_\Theta(\underline{y}) = \frac{1}{Z(\Theta)} P(\underline{y}) B_\Theta(\underline{y}), \tag{24}$$

with the normalization $Z = \sum_{\underline{x}} P(\underline{x}) B_\Theta(\underline{x})$.

From physical intuition, it is easy to understand the effect of the bias depending on the temperature $\Theta$:

- For $\Theta = \infty$ the argument of the exponential is zero. Thus, the bias has no effect and one is back to standard sampling according to $P(\underline{y})$. Thus with respect to $Q(S)$, typical values will be sampled, in the region where the probability $Q(S)$ is large.

- If $\Theta \to 0^+$, the temperature is small but positive. This means, from a physical point of view, ground states with respect to $S$ are sampled, i.e., configurations $\underline{y}$ with the lowest values of $S = S(\underline{y})$. Thus, the left tail of $Q(S)$ is addressed.

- In contrast to the physical canonical ensemble, also temperatures $\Theta < 0$ make sense. Now, the bias is larger for configurations where $S = S(\underline{y})$ is larger than typical values of $S$.

- In particular for $\Theta \to 0^-$, configurations $\underline{y}$ with the largest values of $S$ are preferred. This addresses the right tail of $Q(S)$.

During the generation of configurations $\underline{y}$, one measures always the corresponding values $S = S(\underline{y})$, which finally can be collected in histograms which approximate the distributions $Q_\Theta(S)$ of $S$ in the biased ensemble at temperature $\Theta$. The results will depend of $\Theta$. This means that the histograms will cover a certain range of values of $S$. Therefore, one usually has to generate configurations for several values of $\Theta$ in order to obtain a full or at least a large range of values of $S$. Details on how the final estimate for the distribution $Q(S)$ can be obtained from combining the result generated at different values of $\Theta$ is presented in Sec. 4.3.

The generation of configurations according to the biased distribution $P_\Theta(\underline{y})$ can typically not be performed by direct sampling, i.e. the inversion method cannot be applied. Thus, we use a Markov chain Monte Carlo approach, which is very general, as we have learned in Sec. 3.1. Specifically, we use the versatile Metropolis-Hastings algorithm as introduced in Sec. 3.3. Here, we assume that the Monte Carlo trial moves, as described by the matrix $A(\underline{y} \to \underline{z})$, correspond to the original probabilities $P(\underline{y})$. Thus, if one performed the trial moves and accepted them

all, the original distribution would be obtained. This means, detailed balance holds for this part, i.e. $P(\underline{y})A(\underline{y} \to \underline{z}) = P(\underline{z})A(\underline{z} \to \underline{y})$.

Now, not all trial configurations are accepted, instead a Metropolis criterion is imposed that, as we now will see, depends on the bias. When using Eq. (20) for the Metropolis criterion with the target distribution $P_\Theta(\underline{y})$, and inserting Eqs. (24) and (23), we obtain

$$
\begin{aligned}
\tilde{W}(\underline{y} \to \underline{z}) &\overset{(20)}{=} \min\left(1, \frac{P_\Theta(\underline{z})}{P_\Theta(\underline{y})} \frac{A(\underline{z} \to \underline{y})}{A(\underline{y} \to \underline{z})}\right) \\
&\overset{(24)}{=} \min\left(1, \frac{P(\underline{z})B_\Theta(\underline{z})}{P(\underline{y})B_\Theta(\underline{y})} \frac{P(\underline{y})}{P(\underline{z})}\right) \\
&= \min\left(1, \frac{B_\Theta(\underline{z})}{B_\Theta(\underline{y})}\right) \\
&\overset{(23)}{=} \min\left(1, \exp(-(S(\underline{z}) - S(\underline{y}))/\Theta)\right).
\end{aligned}
\tag{25}
$$

Note that, depending on the model, it is not necessarily the most efficient choice to generate the trial configurations $\underline{z}$ such that the original distribution is reproduced. This choice might lead to low acceptance probabilities of the Metropolis steps. Sometimes it is better to generate trial configurations where one has the biased distribution already in mind. In this case, the Metropolis acceptance probability will look different, depending on the model and the details of the trial-configuration generation.

Nevertheless, generating the trial configurations compatible with the original distribution $P(\underline{y})$ is often beneficial. For this case, the algorithm is summarized here, $n_{\mathrm{mc}}$ being the number of MC steps:

**algorithm** biased-MC($n_{\mathrm{mc}}$, $\Theta$)
**begin**
    generate start configuration $\underline{y}(0)$
    **for** $t = 1, \ldots, n_{\mathrm{mc}}$
    **do**
        generate trial configuration $\underline{z}$ from $\underline{y}(t-1)$ compatible with $P(\underline{z})$
        compute $\Delta S = S(\underline{z}) - S(\underline{y}(t-1))$
        $r$: uniform $U(0,1)$ random number
        **if** $r < \min(1, \exp(-\Delta S/\Theta))$
        **then**
            $\underline{y}(t) := \underline{z}$
        **else**
            $\underline{y}(t) := \underline{y}(t-1)$
    **done**
**end**

In general, how the trial configuration is generated from the current configuration depends a lot on the model, and for most models there are still many ways. Typically, only small changes of the configuration lead to sufficiently large acceptance probabilities. As an example, we now consider again the Bernoulli process.

## 4.2 Bias for the Bernoulli process

For the Bernoulli process of $n$ coin flips, the MCMC algorithm redraws in each step a number $n_{\mathrm{c}}$ of randomly chosen elements of the current configuration $\underline{y}(t)$, with the original coin prob-

ability $\alpha$ as of Eq. (1). For the changed elements, the current values are stored in a vector, i.e. an array `old[]`. This allows to restore the previous configuration, if the trial configuration is rejected. Note that for restoring the configuration, the current values are restored in reverse order, to prevent a mistake if some elements are by chance selected more than once for a change.

We also assume that the quantity $S$ of interest is a general function of the configuration $\underline{y}$. The MCMC algorithm for performing $n_{\text{mc}}$ steps for the Bernoulli process, performed at value $\Theta$ for the temperature-like parameter, reads as follows: It is assumed that the initial or current configuration is passed as vector $\underline{y}$ to the function, all necessary parameters, and the score evaluation function $S(\cdot)$.

**algorithm** biased-MC-Bernoulli($\underline{y}$, $n$, $n_{\text{mc}}$, $n_{\text{c}}$, $\Theta$, function $S(\cdot)$)
**begin**
    **for** $t = 1, \ldots, n_{\text{mc}}$
    **do**
        $S_{\text{old}} = S(\underline{y})$
        **for** $c = 1, \ldots, n_{\text{c}}$
        **do**
            i = random in $\{1, \ldots, n\}$
            old$[c]$ = (i, $y_i$)
            $r$: uniform $U(0,1)$ random number
            **if** $r < \alpha$ **then** $y_i = 1$ **else** $y_i = 0$
        **done**
        $S_{\text{new}} = S(\underline{y})$
        compute $\Delta S = S_{\text{new}} - S_{\text{old}}$
        $r$: uniform $U(0,1)$ random number
        **if** $r \geq \min(1, \exp(-\Delta S/\Theta))$
        **then** "reject"
            **for** $c = n_{\text{c}}, n_{\text{c}} - 1, \ldots, 1$
            **do**
                restore $y_i$ where $i$ and $y_i$ are stored in old$[c]$
            **done**
    **done**
**end**

Note that in case of acceptance, nothing has to be done, since the trial state is stored directly in $\underline{y}$. The C code [2] can be found in `mc_bernoulli.c` as function `bernoulli_mc_step_bias()`.

As an unusual example for the measurable quantity $S$, we consider the number $S_{3+}$ of blocks of at least 3 1's in a row. For example, the configuration $\underline{y} = 010110\underline{11}100\underline{111}101$ has $S_{3+} = 2$.

With the `mc_bernoulli.c` code [2] we can generate data for $S_{3+}$, The Metropolis algorithm is chosen by the `-alg 2` option. First, we simulate the almost unbiased case for which we choose $\Theta = 1000$. For this case, a large number $n_{\text{c}} = 45$ of changes are performed in each step. This is stated with the `-change` option when calling the program. Here, we consider $n = 51$ coin flips for $\alpha = 0.5$, which are the third and second last arguments, respectively. The number of Monte Carlo steps is the last argument, 100000 here. The first 10000 steps are for equilibration and therefore ignored for measuring the histogram, as stated by the argument for the `-histo` option. When calling the program in Unix from a command line, this can be achieved by entering

```
mc_bernoulli -S3+ -alg 2  -histo 10000 -theta 1000 \
-change 45 51 0.5 100000 > bernoulli_S3p_t1000.histo
```

Note that the results are redirected to the file `bernoulli_S3p_t1000.histo`. To next access the upper tail, we choose $\Theta = -0.5$. Here only 5 elements of the coin flip vector are redrawn in each MC step, to allow for a considerable acceptance probability.

```
mc_bernoulli -S3+ -alg 2  -histo 10000 -theta -0.5 \
-change 5 51 0.5 100000  > bernoulli_S3p_t-05.histo
```

The two resulting histograms are shown in Fig. 7. The typical-case sampling is obtained for $\Theta = 1000$, because the bias is almost 1. This results in observed values $S_{3+} \in [0, 8]$. For $\Theta = -0.5$, the bias shifts the observed distribution to larger values up to $S_{3+} = 12$. Note that for $n = 51$ the maximum possible value is $S_{3+} = 13$. Thus, almost the full support of the distribution $Q(S_{3+})$ is sampled.

You should perform simulations yourself using the code and observe how varying the parameters, like the number $n_c$ of redrawn coin flips, lead to changes of the results. In particular the empirical acceptance probability is affected, which is also written into the output file. It is also interesting to change $\Theta$ and try to sample the largest possible values, which is here $S_{3+} = 13$.

Note that the $S_{3+}$ quantity has the property that it does not depend monotonously on the model parameter $\alpha$. If one tried, in the spirit of the educated bias as shown in Sec. 2.4, to simulate the process for parameter values $\beta \neq \alpha$, one would observe:

- If $\beta < \alpha$ is small, i.e., $\beta < \alpha$. there will be fewer 1's in the configuration, thus $S_{3+}$ is often zero, e.g. $\underline{y} = 0000100010100100$ has $S_{3+} = 0$. This could be used to access the left tail.

- To obtain a larger number of $S_{3+}$ one has to use not too-small values of the parameter. For $\beta \approx \alpha$ there might be higher than typical values of $S_{3+}$. E.g., $\underline{y} = 1011101011101110$ exhibits $S_{3+} = 3$. But this will occur with the typical frequency, i.e. not so frequently.

- For $\beta \gg \alpha$, one obtains again lower than typical values of $S_{3+}$, e.g., $\underline{y} = 1111111011111111$ has $S_{3+} = 2$.

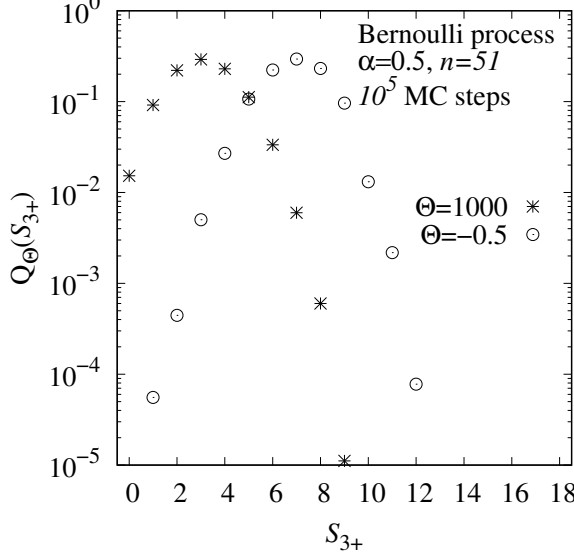

Figure 7: Histograms of biased MCMC sampling of $S_{3+}$ for the Bernoulli process case ($n = 51$ coin flips at $\alpha = 0.5$). Two cases of the bias temperature $\Theta = 1000$ and $\Theta = -0.5$ were considered.

Thus, this quantity is an example where the *educated* approach, i.e. just using a different value $\beta$ for the Bernoulli parameter, will not allow one to reach the very upper tails of the distribution $Q(S_{3+})$. Only a *blind* approach, which takes automatically care of how the configuration space influences the measured values, will work.

Still, one has to take some care, because the rare configurations with untypical small values of $S_{3+}$ will consist of sequences of $n$ coin flips with unusual small or unusual high number of 1's. For the present example, where the system size $n = 51$ is not very large, this poses no problem. But for much larger system sizes $n$, it will be more difficult to obtain the correct result. Here, a single MCMC will not grant access to both regions within one run, because the two regions are separated by a region where $S_{3+}$ is typically large. Therefore, this region acts as a barrier in configuration space. This can be improved by performing many runs each time staring in the region of typical configurations. Even better would be to use an approach like *Parallel Tempering* [12, 13]. It is based on performing the simulations for several temperature parameters $\Theta_1, \ldots, \Theta_K$ in parallel, while allowing the configurations to switch temperatures in a controlled way, also governed by a Metropolis criterion. This approach is suited to explore broad ranges of the configurations space.

As final example, we show a measurable quantity, where it is even clearer that the *educated* approach will not work. We consider the number $S_{0101}$ of non-overlapping segments 0101 in $\underline{y}$. The following two example sequences have exactly half of the entries 1, and the other half 0:

- $\underline{y} = 0000000011111111$ has $S_{0101} = 0$,

- $\underline{y} = 0101010101010101$ has $S_{0101} = 4$,

while the resulting values for $S_{0101}$ are very different. Here it is even more obvious that the *educated* approach of performing simulations for other values $\beta \neq \alpha$ of the model parameter will not give access to the tails of the distribution. Here, only the more general *blind* approach, which depends on the value $S$ of interest, is suitable.

## 4.3 Obtaining the true distribution

The final step is to obtain for the quantity $S$ of interest the estimate of the original distribution $Q(S)$, see Eq. (22). Since the sum over the exponential many configurations cannot be performed, $Q(S)$ shall be estimated from the simulations. The starting point for the analysis is the collection of histograms estimating the distributions $Q_\Theta(S)$ in the biased ensemble, for different values of the temperature-like parameter $\Theta$. One has

$$
\begin{aligned}
Q_\Theta(S) &= \sum_{\underline{y}} \delta_{S,S(\underline{y})} P_\Theta(\underline{y}) \\
&= \sum_{\underline{y}} \delta_{S,S(\underline{y})} \frac{1}{Z(\theta)} P(\underline{y}) B_\Theta(\underline{y}) \\
&= \frac{1}{Z(\Theta)} \sum_{\underline{y}} \delta_{S,S(\underline{y})} P(\underline{y}) \exp(-S(\underline{y})/\Theta) \\
&\stackrel{(\star)}{=} \frac{1}{Z(\Theta)} \exp(-S/\Theta) \sum_{\underline{y}} \delta_{S,S(\underline{y})} P(\underline{y}) \\
&= \frac{1}{Z(\Theta)} \exp(-S/\Theta) Q(S) \\
\Rightarrow \quad Q(S) &= Z(\Theta) \exp(S/\Theta) Q_\Theta(S).
\end{aligned}
\tag{26}
$$

For equality ($\star$), due to the delta factor $\delta_{S,S(\underline{y})}$, we have $\exp(-S(\underline{y})/\Theta)$ is equal to $\exp(-S/\Theta)$. Therefore, it can be moved in front of the sum. The results means that the true distribution $Q(S)$ can be obtained from the biased distribution $Q_\Theta(S)$! Note that the right side depends on $\Theta$, while the left does not. Thus, theoretically, the data obtained for any value of $\Theta$ is sufficient to obtain $Q(S)$. In practice, the accumulated data for any value of $\Theta$ will concentrate in some interval $I^\Theta = [S^\Theta_{\min}, S^\Theta_{\max}]$, thus $Q(S)$ can be estimated from $Q_\Theta(S)$ only in this interval. For practical reasons, we assume that this interval consist only of values where the statistics is "good enough", i.e. does not contain rare outliers with respect to $Q_\Theta(S)$. Since such an interval is finite, to cover a large range of the support of $Q(S)$, one has to perform simulations for several values of $\Theta$.

Also it is interesting that one has to divide in Eq. (26) by $\exp(-S/\Theta)/Z(\Theta)$ which corresponds to $B_\Theta(\underline{y})$. This is similar to unbiasing in Eq. (6). The only difference is that previously it was assumed that the bias is fully known, while here the values $Z(\Theta)$ are not known a priori. Next, we will discuss how these normalization constants can be obtained.

The basic idea goes as follows: Assume that for two values $\Theta_1$, $\Theta_2$ the intervals $I^{\Theta_1}$ and $I^{\Theta_2}$ of sufficiently sampled data overlap, i.e. $I = I^{\Theta_1} \cap I^{\Theta_2}$ is not empty. Thus, for $S \in I$, we have substantial data to estimate both $Q_{\Theta_1}(S)$ and $Q_{\Theta_2}(S)$. Here one has

$$Z(\Theta_1)\exp(S/\Theta_1)Q_{\Theta_1}(S) = Q(S) = Z(\Theta_2)\exp(S/\Theta_2)Q_{\Theta_2}(S). \tag{27}$$

Thus, the ratio of the normalization constants is fixed to

$$\frac{Z(\Theta_1)}{Z(\Theta_2)} = \frac{\exp(S/\Theta_2)Q_{\Theta_2}(S)}{\exp(S/\Theta_1)Q_{\Theta_1}(S)}. \tag{28}$$

Since the sampled and normalized histograms only approximate the distributions, this relation will also be fulfilled only approximately. Note that for $K$ different values $\Theta_1, \ldots, \Theta_K$ one can determine $K-1$ ratios. In addition, the final estimate for $Q(S)$ should be normalized as well. This can be approximated by $Z(\infty) = 1$ and provides the final relation to fix all normalization constants.

In order to actually arrive with the given histograms at the estimate for the final distribution $Q(S)$, we first discuss a manual approach. We restrict ourselves to two temperatures $\Theta_1$ and $\Theta_2$, the generalization to more temperatures is straightforward. This approach means, we choose $Z(\Theta_1)$ and $Z(\Theta_2)$ such that when plotting the two rescaled histograms they almost agree, within statistical fluctuations, in the interval $I$ of values of $S$ where both histograms have gathered sufficiently data. One can say, the distributions are "glued" or "stiched" together.

We show how this works for the data as generated in the last section, using the `gnuplot` program. For the data obtained at $\Theta_1 = 1000$, which is effectively infinity, we assume $Z(\Theta_1) \approx 1$. To start the manual adjustment of $Z(\Theta_2)$, we also start with $Z(\Theta_2) = 1$ and plot the histograms rescaled by the exponential bias with the following `gnuplot` command:

```
gnuplot> Z1=1
gnuplot> Z2=1
gnuplot> plot [-0.5:18] \
   "bernoulli_S3p_t1000.histo", using 1:(Z1*exp($1/1000)*$2) \
   "bernoulli_S3p_t-05.histo" using 1:(Z2*exp($1/-0.5)*$2)
```

Note that we have used the `using` data modifier, which can be abbreviated by the letter u. The data modifier `1:(Z1*exp($1/1000)*$2)` of the first plotted curve, provided by the file `"bernoulli_S3p_t1000.histo"`, states that the $x$ coordinate is just taken from the first column, while the $y$ coordinate is the product of the value for Z1, times the exponential of the value in the first column ($1) devided by the temperature 1000, times the value in the second column ($2). The result will look like Fig. 8.

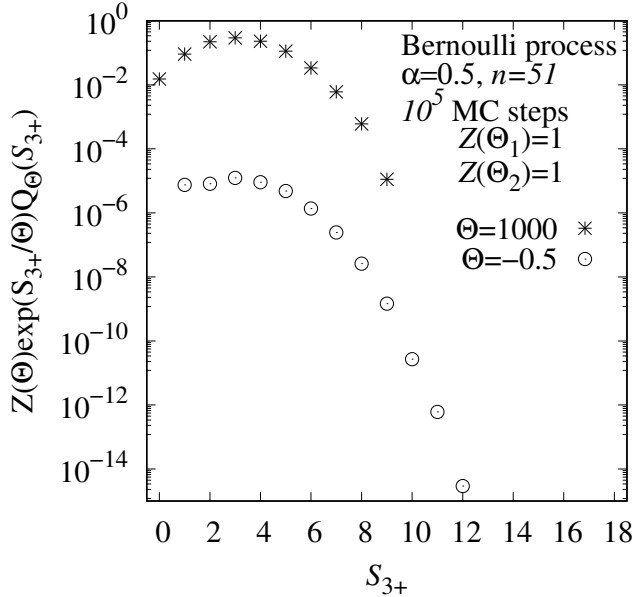

Figure 8: Rescaled histograms of biased MCMC sampling of $S_{3+}$ for the Bernoulli process ($n = 51$ coin flips at $\alpha = 0.5$). Two cases of the bias temperature $\Theta = 1000$ and $\Theta = -0.5$ were considered. The initial values for the partition functions are $Z(1000) = 1$ and $Z(-0.5) = 1$.

As one can see, the shape of the rescaled histogram for $\Theta = -0.5$ looks similar to the histogram for $\Theta = 1000$, but the frequencies for $\Theta = -0.5$ are some orders of magnitude too small. This means $Z(-0.5)$ must be larger, about a factor of $10^4$. Now one can iteratively refine $Z(-0.5)$ and replot, until the two rescaled histograms agree well in the overlapping region. Here one could be satisfied with $Z(-0.5) = 2.3 \times 10^4$, i.e.

```
gnuplot> Z2=2.3e4
gnuplot> plot [-0.5:18] \
   "bernoulli_S3p_t1000.histo", u 1:(Z1*exp($1/1000)*$2) \
   "bernoulli_S3p_t-05.histo" u 1:(Z2*exp($1/-0.5)*$2)
```

The result will look like Fig. 9. Note that the histogram for $\Theta = 1000$ is actually not much rescaled since still $Z(1000) = 1$ and $\exp(-S_{3+}/1000)$ is also about 1. Now the two rescaled histograms agree well in the middle of the overlapping region, say $S \in [3, 8]$. One could assemble the final histogram, which is not shown here, by taking each data point from the biased and rescaled histogram where the statistics is best. For the present example this would be for $S \in [0, 5]$ from the $\Theta = 1000$ result and for $S \in [6, 12]$ from the $\Theta = -0.5$ result.

Note that for a larger system size $n$, one needs a larger number of temperatures. In this case one could, after gluing the results for $\Theta = 1000$ and $\Theta = -0.5$, rescale the third histogram, say obtained at $\Theta = -0.2$ for even larger values of $S_{3+}$, such that it matches the so far estimated distribution. For very large systems of other models, one might have to perform simulations for a number $K$ of temperatures up to several hundreds [14, 15]. Here, the calculation of all partition functions and obtaining the final distribution by hand would be tedious. Thus, an automated approach is better suited. This is also the case for few temperatures, because of the higher precision.

One way to numerically rescale two histograms obtained at neighboring temperatures $\Theta_k, \Theta_{k+1}$ for $k = 1 \ldots, K - 1$ such that they mostly agree, is to minimize the mean-squared difference between the rescaled histograms in the overlapping region $S \in I^{k,k+1}$. Here we assume that $Z(\Theta_k)$ is already known; it could be $Z(\infty) \approx 1$, or obtained from rescaling of other

parts of the distribution. When writing $r_\Theta(S) = \exp(S/\Theta)\tilde{Q}_\Theta(S)$, where $\tilde{Q}_\Theta(S)$ is the actually obtained normalized histogram at temperature $\Theta$, the minimization with respect to $Z(\Theta_{k+1})$ translates into,

$$
\Delta \equiv \sum_{S \in I^{k,k+1}} \left( Z(\Theta_k)r_{\Theta_k}(S) - Z(\Theta_{k+1})r_{\Theta_{k+1}}(S) \right)^2 \to \min
$$

$$
\Rightarrow \quad 0 = \frac{d\Delta}{dZ(\Theta_{k+1})} = \sum_{S \in I^{k,k+1}} 2r_{\Theta_{k+1}}(S)\left( Z(\Theta_k)r_{\Theta_k}(S) - Z(\Theta_{k+1})r_{\Theta_{k+1}}(S) \right)
$$

$$
\to \quad Z(\Theta_{k+1}) = \frac{Z(\Theta_k)\sum_{S \in I^{k,k+1}} r_{\Theta_k}(S)r_{\Theta_{k+1}}(S)}{\sum_{S \in I^{k,k+1}} r_{\Theta_{k+1}}(S)^2}. \tag{29}
$$

It is assumed that the histograms are binned, therefore the sums $\sum_{S \in I^{k,k+1}}$ run over the corresponding bins in the overlapping region. As above, one would choose the intervals $I^{k,k+1}$ such that they contain enough data, for both histograms.

One could start with a histogram obtained by unbiased sampling and assume $Z(\infty) = 1$ there. Then one can obtain the partition functions for neighboring values of $\Theta$, i.e. where $|\Theta|$ is rather large, and so on.

In the version above, all contributions to the sums are equally weighted. One could also use, in different ways, weights which represent the combined qualities of the statistics for the different bins $S$. A high quality means that both histograms for $\Theta_k$ and $\Theta_{k+1}$ contain many entries. Correspondingly, weights could be used to assemble the final result $Q(S)$ from all histograms, such that for each value of $S$ two or possibly even more histograms contribute. For a simple approach, which is usually sufficient, one takes only the histograms that have the best statistics for a given bin.

For completeness, one can evaluate the normalization factor, i.e., the integral, of the final result and divide by it to get a final normalization of exactly one.

There are alternatives for obtaining the normalization constants. One could, e.g. calculate relative normalization constants $Z(\Theta_k)/Z(\Theta_{k+1})$ directly using Eq. (27) for single bin values

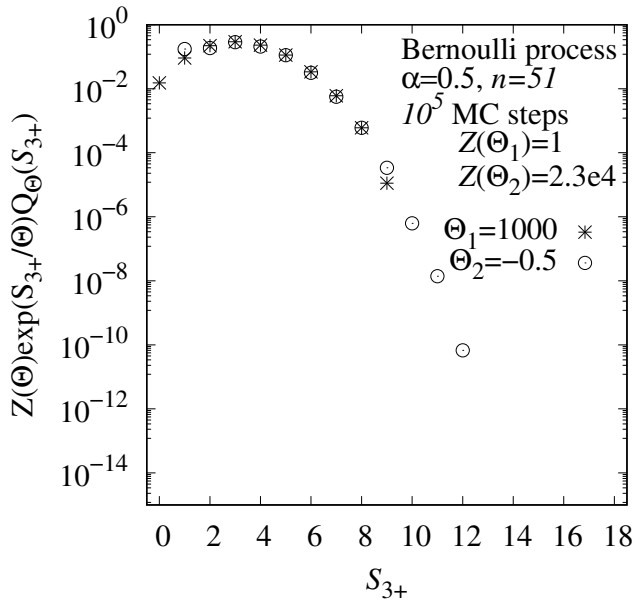

Figure 9: Rescaled and glued histograms of biased MCMC sampling of $S_{3+}$ for Bernoulli process case ($n = 51$ coin flips at $\alpha = 0.5$). Two cases of the bias temperature $\Theta_1 = 1000$ and $\Theta_2 = -0.5$ were considered. The final values for the partition functions are $Z(1000) = 1$ and $Z(-0.5) = 2.3 \times 10^{-4}$.

$S \in I^{k,k+1}$ directly. This gives a certain number of estimates for the ratio of the normalization constants. The final result for $\Theta_k$ and $\Theta_{k+1}$ is then obtained by averaging over these estimates, possibly weighted by the accuracy of the estimates, i.e., by the statistics of the contributing bins.

Even better would be to apply the multi-histogram reweighting approach from Ferrenberg and Swendsen [16]. It includes precision/statistics of different histogram entries and therefore allows for calculation of error bars. Note that you do not have to implement the approach yourself, because there is a Python-based tool by Peter Werner [17] which does the job for you.

## 4.4 Generalization and other examples

For the so-far used example, the Bernoulli process with parameters probability $\alpha$ and size $n$, the state of the Markov chain was given by a vector $\underline{y} \in \{0,1\}^n$. From the given vector at MC time $t$ the quantity of interest $S(t) = S(\underline{y}(t))$ was calculated. Along the Markov chain $\underline{y}(0) \to \underline{y}(1) \to \dots$ typically some entries of the state changed from time step $t$ to time $t+1$, with corresponding changes of the quantity $S$. This can be depicted as follows:

$$
\boxed{\underline{y}(t)} \longrightarrow \boxed{\underline{y}(t+1)} \longrightarrow
$$
$$
\Downarrow \qquad\qquad \Downarrow
$$
$$
S(t) \qquad\qquad S(t+1)
$$
,

where the $\longrightarrow$ symbol indicates the random evolution of the states while the $\Downarrow$ symbol represents a deterministic evaluation.

Within the Metropolis algorithm a trial state $\underline{z}$ is obtained by first copying the current state $\underline{y}(t)$ and then changing some randomly chosen entries $z_i$. The new entries are drawn with the original statistics, i.e., each time a uniformly $U(0,1)$ distributed random number $\xi$ is drawn, and then $z_i = 1$ if $r < \alpha$ and $z_i = 0$ otherwise.

Note, it would be completely equivalent, if instead of storing the result $y_i$ of a coin flip, we store the random numbers used to generate the coin flip in a vector $\underline{\xi} \in [0,1]^n$, i.e. $\xi_i$ is the random number for the $i$'th coin flip. Then one could deterministically assign the coin-flip vector $\underline{y} = \underline{y}(\underline{\xi})$ by $y_i = 1$ if $\xi_i < \alpha$ and $y_i = 0$ else. Thus, instead of evolving a Markov chain of the coin flip vectors $\underline{y}(t)$, from which $S = S(\underline{y}(t))$ one could evolve a Markov chain of random vectors $\underline{\xi}(t)$ and obtain $S = S(\underline{y}(\underline{\xi}(t)))$, i.e. $S = S(\underline{\xi}(t))$, with formally a different function dependency of $S$ on $\underline{\xi}$ as compared to $S$ on $\underline{y}$, here denoted for simplicity both by $S(\cdot)$. The modified approach can be depicted as follows:

$$
\boxed{\underline{\xi}(t)} \longrightarrow \boxed{\underline{\xi}(t+1)} \longrightarrow
$$
$$
\Downarrow \qquad\qquad \Downarrow
$$
$$
\underline{y}(t) \qquad\qquad \underline{y}(t+1)
$$
$$
\Downarrow \qquad\qquad \Downarrow
$$
$$
S(t) \qquad\qquad S(t+1)
$$
.

Clearly, both approaches, having a Markov chain of states $\underline{y}(t)$, and having a Markov chain of states $\underline{\xi}(t)$, are equivalent, only that the latter one looks more complicated. Nevertheless, the latter one is more general! The reason is that whatever stochastic model you investigate, and whatever random quantities you calculate within the model, the implementation will always be based on $U(0,1)$ uniformly distributed random numbers. Some of these numbers may, depending on the model, be turned by the inversion method to exponentially distributed numbers, some by the Box-Muller approach to Gaussian numbers, some random numbers may be used to decide the random orientation of some spins, or the random positions of nodes in a

plane, and so on. This means, basically the implementation of any random process can be represented as a, often complex, deterministic mapping from a vector of $n$ uniformly distributed random numbers to a state of the system. Then, from the state some quantity $S$ of interest can be calculated, in a more or less complex way. Thus, for any random process, the quantity of interest is a deterministic function $S = S(\underline{\xi}(t))$ of a vector $\underline{\xi}$ of random numbers. On the implementation side, this means one disentangles the generation of the random numbers from the other ingredients of a model. Normally, one would call a random number generator at different places in the code. With the disentangles approach, one would first generate all $U(0,1)$ distributed (pseudo) random numbers, store them in the vector and use them when needed, while the other parts of the simulation are performed.

In this way, one can still easily implement direct sampling, by first randomly choosing all entries of $\underline{\xi}$ uniformly in $[0,1]$ and then evaluating $S$. The histogram of observed values of $S$ will give an estimate of the underlying distribution $P(S)$ in the high-probability region. On the other hand, if one is interested in the tails of $P(S)$ one can introduce a bias, say proportional to $\exp(-S(\underline{\xi})/\Theta)$ with parameter $\Theta$, and then use a Markov chain Monte Carlo simulation to drive the simulation to the rare values of $S$, as it has been demonstrated so far. This means, the large-deviation approach presented here is very general and can be applied, at least in principle, to any stochastic system, equilibrium and non-equilibrium ones, from various fields of science.

Consequently, the large-deviation behavior of a large variety of models has been studied with this approach.

### 4.4.1 Random walk

A simple example is the random walk, which is just the sum of, typically independent, random numbers $y_i$ called *increments*. The increments can have arbitrary distribution, e.g., discrete, Gaussian, exponential or power-law. Thus, one can generate a $n$-step random walk from a given vector $\underline{\xi}$ of uniformly $U(0,1)$ distributed random numbers by generating random numbers $y_i$ from the desired distribution and then summing. The resulting position at time $t \le n$ is given by $x(t) = \sum_{i=1}^{t} y_i$. Standard random walks are well understood, i.e. $P(x)$ is well known. More interesting are extensions, e.g., when the increments are correlated. One special variant, with a power-law correlation, is called *fractional Brownian motion* [18]. Here, depending on the so called *Hurst* exponent $H$, the increments are positively ($H > 1/2$), negatively ($H < 1/2$) or not ($H = 1/2$) correlated. The correlation can be conveniently imposed [19,20] by using a Fourier Transform (FT) of the correlation function plus a back transformation of the product of the randomness and the FT correlation. This process becomes even more challenging when one introduces an absorbing boundary at, say, $x = 0$. This means that only walks are considered where for all times $t$ the position $x(t)$ is not negative. Here, the distribution $P(x)$ for $x = x(n)$ is of interest near $x = 0$, since here the probabilities are very small, due to the absorbing boundary. Unfortunately, the direct sampling approach will generate many walks where at some time $x(t) < 0$ and thus one has to disregard the walk. The probability of not being disregarded, the so-called *persistence*, decreases like a power law $\sim t^{H-1}$. Here, a MCMC approach comes handy even without a bias, because one can restrict the Markov chain to walks which do not get absorbed. This can be achieved within the Metropolis algorithm by rejecting any trial configuration, where the resulting walk $x(t)$ shows an absorption. By using in addition also a bias $\sim \exp(-x(n)/\Theta)$, the low probability tail of $P(x)$ has been determined successfully within a large-deviation study [21]. This allowed the authors to numerically verify that indeed $P(x)$ follows a power law $\sim x^{\phi}$ near $x = 0$, where the power $\phi$ depends on the Hurst exponent like $\phi = (1-H)/H$ as predicted analytically [22,23].

### 4.4.2   Non-interacting fermions

Also simple by its definition, but slightly more involved with respect to the large-deviation approach, is the following model. It describes the ground state of $K$ non-interacting fermions in a random energy landscape [24] of $n$ sites. The potential landscape is just a vector $\underline{y}$ of $n$ independently and identically distributed random numbers $y_i$ following an arbitrary given distribution, e.g. exponential. Thus, a realization of the landscape can again be generated easily from a vector $\underline{\xi}$ of $n$ uniform $U(0,1)$ random numbers. The fermions occupy the $K$ lowest energy levels, thus the ground state energy $E_0$ is obtained by sorting the energies in $\underline{y}$ and summing up the lowest $K$ values. The tails of the distribution $P(E_0)$ have also been addressed [25] by a large-deviation approach with a bias $\sim \exp(-E_0/\Theta)$. Here, the difficulty arises that $E_0$ depends a lot on the actual entries in $\underline{\xi}$. The far tails are governed by vectors $\underline{\xi}$ where at least $K$ entries are all very close to 0 or all very close to 1. Here, redrawing entries $\xi_i$ completely in $[0,1]$ would change the resulting value $E_0$ too much, leading to too many rejections within the Metropolis algorithm, i.e. no progress. Instead an approach was used, where for the generation of the trial states one starts by selecting a random magnitude $\delta$ among six values $\delta \in \{10^0, 10^{-1}, \ldots, 10^{-5}\}$. Then some randomly chosen entries $\xi_i$ are changed only a bit by applying $\xi_i = \xi_i + \epsilon\delta$, where $\epsilon$ is uniformly $U(-1,1)$ distributed. Moves leading $\xi_i$ outside the interval $[0,1]$ lead also to an immediate rejection of the change of the entry $\xi$. Thus, the entries $\xi_i$ perform random walks within the interval $[0,1]$ with reflecting boundary conditions, which guarantees a uniform distribution in $[0,1]$. If the actual composition of $\underline{\xi}$ is crucial, the moves where $\delta$ is too large lead to too large changes of $E_0$ and are therefore rejected. But there will be always trial-state generations where $\delta = 10^{-4}$ or $\delta = 10^{-5}$. These are typically accepted and lead to a slow but steady evolution of the states within the Markov chain. By using this approach, the distribution $P(E_0)$ of the ground states energies could be determined down to probabilities as small as $10^{-160}$.

### 4.4.3   Traffic model

A more complex model is, e.g. the Nagel-Schreckenberg model [26]. It describes a one-lane one-directional traffic of cars, where the density $\rho$ of cars is the control parameter. The cars accelerate in each time step a bit if possible or break if they come too close to cars in front of them. In addition to these deterministic rules, there is some random breaking, even if there is enough space. For these random breaking events, collected along a full temporal development, a vector $\underline{\xi}$ of random number is used, which defines the complete randomness of the dynamics. The distribution $P(q)$ of the traffic flow, measured at the end of the dynamical evolution over some time, was investigated. By considering an exponential bias $\sim \exp(-q/\Theta)$, performing a Markov chain in which the randomness $\underline{\xi}$ represents the state, and subsequent unbiasing $\sim \exp(+q/\Theta)$, the distribution $P(q)$ could be determined [27] down to probability densities such as $10^{-160}$. In particular it was observed that the shape of the tails of $P(q)$ change, depending on the density $\rho$, whether the system is in the low-density high-flow phase, or in the medium or high-density congested phase.

### 4.4.4   Spread of diseases

Also many models of the evolution of the spread of a diseases on networks feature stochastic dynamics. The nodes of the network represent individuals and the edges personal contacts. For the well known *susceptible-infected-recovered* (SIR) model [28], each node can be in one of these three states. An infected node transfers the disease with some rate or probability $\gamma$ to neighboring nodes that are susceptible. In addition, infected nodes can recover with another rate or probability $\beta$. Typically, the dynamics starts with one or few infected nodes, and stops

when no infected nodes remain. The randomness, to decide whether nodes switch state, i.e. all random ingredients of a fully dynamical evolution, can also be stored in a vector $\xi$. This allows for a biased Marcov chain simulation. Using this approach, large-deviation studies [29] have been performed in particular to investigate the distributions $P(I)$ of the number $I$ of nodes that caught the infection for some time. Not only the distribution $P(I)$ itself was of interest, but also what type of dynamical evolution lead to particular rare but severe outbreaks. Also the influence of vaccinations [30], the effect of lock downs [31], or how wearing masks [32] may limit the spread of the disease have been studied within large-deviation frameworks.

### 4.4.5 Stochastic thermodynamics

Also of interest are processes involving the generation of physical work. They have been increasingly studied in the context of *stochastic thermodynamics*. One setup of interest is where one starts a system coupled to a heat bath held at a physical temperature $T$ in equilibrium. Then one changes, quickly or slowly, an external parameter $B : B_0 \rightarrow B_1$ while the system is still in contact with the heat bath. Changing $B$ leads to some work $W$ being generated. When $B = B_1$ the process ends in a non-equilibrium state of the system. Interestingly, by the equations of Jarzynski [33] and Crooks [34] one is able to obtain from the distribution $P(W)$ of work the free energy difference $\Delta F$ between the states at $B = B_0$ and $B = B_1$. Note that $\Delta F$ is an equilibrium quantity, while the process and $P(W)$ are non-equilibrium. Importantly, in particular if the system consist of more than few particles, one needs to know $P(W)$ down to the very low probability tails in order to determine $\Delta F$. Consequently, for numerical simulations a large-deviation algorithm should be used. These processes typically have several random contributions: the sampling of the initial equilibrium state and the thermal noise which appears while the non-equilibrium process is performed. Thus, the mapping from the vector $\xi$ to the final result $W$ may be quite involved and may require a substential amount of simulation time. This time has to be spent for each single step of the Markov chain which evolves the randomness vector $\xi$. The approach has been applied to non-equilibrium work processes of the Ising model in an external field [35] and the unfolding of RNAs by an external force [36]. In both cases $P(W)$ could be obtained down to sufficiently small probability densities like $10^{-100}$ and the equations of Jarzynski [33] and Crooks [34] could be confirmed.

### 4.4.6 Direct encoding of configurations

Thus, using a vector $\xi$ of random numbers to separate the randomness from the deterministic evolution of a system is very convenient. Nevertheless, there are still many examples where the large-deviation properties of a system have been studied by performing a Markov chain with the physical systems being directly used as the state of the chain. Examples are the

- distribution of sequence-alignment scores [37, 38],

- properties of random graphs [39–41],

- stability of steady-state energy grids [42, 43],

- distribution of the length and entropy of longest increasing subsequences for permutations of random numbers [44, 45],

- the distribution of the free energy of directed polymers in disordered media [14, 15, 46]

- ground state energy of random magnets [47].

Many more systems can be studied with respect to large-deviations properties. Let us hope that this introduction has equipped you with the foundations allowing you to enter the field sucessfully!

## 4.5   Computational resources

One might wonder what computational resources are needed to study the tails of the distribution. First, most of the time is spent in running single instances of the model of interest, for a given set of random numbers, or for a given directly encoded instance. This sets the time scale needed to perform the MCMC simulations, say for an exponential bias $\exp(-S/\Theta)$ with a given temperature $\Theta$. A typical number of MC steps is $O(10^6)$. Often a run for a single instance requires only few milliseconds, if order of 1000 variables describe the instance. This means the biased MCMC will run few 1000s, which can be done on a laptop, as it is the case for the simple Bernoulli process considered here. If one single runs requires one second, the MCMC will take correspondingly more.

Second, one has to take into account how many of temperature values $\Theta$ have to be considered, since the simulations have to be performed independently for all of them. Typically, to reach probabilities as small as $10^{-50}$, of the order of 10 temperatures are needed. If one wants to go much lower, many more temperatures are needed. For example in the study of the directed polymers in disordered media [14, 15, 46], probabilities as small as $10^{-1000}$ were obtained, which required more than 200 different values of $\Theta$. Since also the system sizes were rather large, leading to longer running times to evaluate single instances, a parallel implementation on a high-performance cluster was used and required several weeks while running on few hundreds of cores in parallel.

## 4.6   Other approaches

The presented approach of changing random numbers and using an exponential bias is rather general but not completely universal, so other approaches exist.

Sometimes, in particular if $P(S)$ is not concave, the exponential bias $\exp(-S/\Theta)$ leads to an effective distribution $P(S)\exp(-S/\Theta)/Z(\Theta)$ which is not bell-like around a typical value as shown in Fig. 7, but instead could exhibit a two-peak structure [39]. In this case the values of $S$ between the two peaks are hard to access. This might turn even impossible if the system size is large. One way out could be to use a more concentrated bias, which forces the biased distribution to be bell-like. One could, e.g., use a Gaussian bias [48] $\exp(-(S-S_0)^2/(2\sigma^2))$, as it has recently been applied to study the distribution of susceptibilities for diluted magnetic systems [49].

An even more general approach is to aim at using $1/P(S)$ as the bias. Since the resulting distribution is proprtional to $P(S)/P(S)=1$, i.e., one obtains a more or less uniform sampling. This approach is called *Umbrella sampling* [50]. Nevertheless, typically $P(S)$ is now known. In this case one tries to construct $P(S)$ iteratively with increasing accurcy during the simulation, starting from a flat distribution. For this purposes methods like the *Wang-Landau* approach are useful [51], which was applied to study rare trajectories of infection dynamics [29–32], as already mentioned above. A related algorithm is the *Multi-canonical Ensemble* [52] which has been applied for the rare-event case to testing error-correcting codes [53]. Another issue could be that the system of interest is simulated with a given package, which is hard or even impossible to change, such that one cannot replace the internal calls to a random number generator by accesses to a vector of random numbers. Also, sometimes the actual dynamics is free of randomness, i.e., deterministic. In both cases, one could perform small random changes to the initial or intermediate states of the system under consideration. Such approaches are used for rare-event sampling of climate models [54]. Here, the single runs of the original model are so demanding that indeed a MCMC sampling with each step being a full run of the climate model over the full period of interest would require too much time. Here, in particular in the case where the quantity $S$ of interest is integrated over time., e.g., the physical temperature in the climate model, population based approaches, so called *Cloning Algorithms*

[55], sometimes referred as genetic selection algorithms, are convenient. Here, instances with slightly different initial conditions are simulated, and after a suitable number of steps for the model, the population is reshaped according the biased probabilities of the different population members. Such approaches have also been used for some models, like the *Totally Asymmetric Exclusion Process* (TASEP) [56].

For other problems, sometime one is not interested in particular rare endpoints of a process, but in rare trajectories between known endpoints. This is the case when considering configurational changes of proteins. Here the *Transition-Path Sampling Approach* [57,58] is very useful.

## 5 Conclusion

This text provides the fundamentals to access the tails of distributions in simulations of stochastic systems. Then main idea of the central approach is to separate the randomness from the other ingredients of the model and keep the randomness in a vector of $U(0,1)$ distributed entries. This allows one to perform a Markov Chain Monte Carlo simulation including a bias, which drives the simulation to the desired range of values. Particularly useful is the exponential bias, with a temperature-like parameter controlling the range of values. By performing simulation for different temperatures, often the distribution can be obtained numerically over hundreds of decades in probability. Here, the methods are illustrated using a very simple model, the Bernoulli process. Since any stochastic simulation relies in principle on $U(0,1)$ random numbers, the approach is very general and can be applied to a wide range of equilibrium and non-equilibrium models. Still, many issues can arise, like choosing a suitable bias or to determine efficient moves when changing the configurations in the Markov chain.

## Acknowledgments

The author thanks Abishek Dhar, Joachim Krug, Satya N. Majumdar, Alberto Rosso, and Grégory Schehr for organizing the summer school "Theory of Large Deviations and Applications" in Les Houches 2024 and for inviting him to give a series of lectures which was the basis of the current text. The author is thankful to Peter Werner for critically reading the manuscript. He furthermore is grateful to Winfried G. Schneeweiß for introducing him to the method of biased simulations which happened during the author's Diploma thesis in Computer Science at the Fernuniversität Hagen (Germany) in 1992. He also thanks Peter Grassberger for introducing him to the field of the statistics of biological sequence alignment during a workshop at Santa Barbara (USA) in 2001, which motivated the first physics large-deviation project [37] of the author. The author finally is much obliged to all his collaborators in the field, which have become too many to list them all here.

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
