# Peer review of "Numerical Aspects of Large Deviations"

_SciPost Physics Lecture Notes, doi:SciPost Phys. Lect. Notes 100 (2025)_

## Round 1 · Referee Report · Anonymous (Referee 1) · 2025-1-9

Report

These much-needed lecture notes provide a comprehensive overview of numerical techniques for probing large-deviation events in stochastic processes. Using the simple yet illustrative example of a Bernoulli process throughout, the author introduces a range of methodologies in a clear and pedagogical manner. The lecture notes effectively cover fundamental concepts such as direct sampling, Markov Chain Monte Carlo, and importance sampling simulations. The inclusion of practical examples and detailed explanations enhances their accessibility, making them a useful resource for both students and researchers.

In the concluding section, the author outlines various applications of these techniques, highlighting their relevance across statistical physics and large deviation theory. Given the scarcity of such lecture notes on this topic, these materials fill an important gap in the literature. I therefore recommend publication, subject to the minor comments listed below.

Requested changes

Minor Comments: 1. Equation (12): The derivation or further justification of this equation is necessary. While it is intuitive in the two extreme cases mentioned, a general explanation is currently missing. 2. After Equation (13): The phrase "later on in section" should be revised to "later in section." 3. After Equation (20): The statement "Hence, the MH choice is maximal!" requires clarification. The author should elaborate on what "maximal" means in this context. 4. Page 17, third paragraph: The sentence "Details how the final... is presented..." should be corrected to "Details on how the final... are presented."

Recommendation

Publish (easily meets expectations and criteria for this Journal; among top 50%)

---

## Round 1 · Referee Report · Anonymous (Referee 2) · 2025-1-12

Report

Here is my report for the manuscript entitled "Numerical Aspects of Large Deviations", submitted to SciPost lecture notes. The paper gives a pedagogical introduction to the topic of simulating rare events in stochastic systems, using the importance sampling technique. Beginning from the most fundamental and standard simulation methods described in sections 2 and 3, the paper gradually builds all the necessary tools and then in section 4, the importance sampling method is described. Throughout the paper, the very simple (but also very pedagogical) example of the Bernoulli process is given, and then towards the end of section 4, several other highly nontrivial examples from the existing literature are briefly discussed.

The topic is of broad interest and the paper is excellently written and does a great service to the statistical physics community, serving both as a useful reference text and also as lecture notes that I believe will be used by many lecturers when teaching courses about large deviations. The author does a great job of explaining a nontrivial topic in simple terms and without assuming too much background knowledge from the reader. I would expect that any graduate student in physics would be able to understand these notes, which is quite an achievement from the point of view of the author. The text is already polished and I recommend its publication in SciPost almost as is: I give quite a few comments but all of them are very minor.

Comments:

  • Bearing in mind that the text will be used as a reference (and not just as actual lecture notes), some readers might have prior knowledge of some of the topics. For instance, I would expect many young readers to know the basic theory of numerical simulations, but nothing about large deviations. E.g., they might know how to simulate Glauber dynamics of the Ising model using the Metropolis algorithm (this is sometimes taught already at the undergraduate level), but they will not know about biasing. So perhaps it would be a good idea to add, at the end of section 1, a paragraph describing the structure of the rest of the paper, and indicate which sections may be skipped (or read very quickly) by readers with such prior knowledge.

  • Typo: "In order the address" --> In order to address

  • Typo: "10^4, 10^6, and 10^6" (appears twice) --> 10^4, 10^6, and 10^8 (?)

  • The author cites Refs. [7,8] in the context of theoretical calculations in large deviation theory. These are of course standard texts that should indeed be cited, however I believe that they might require from the reader a rather large effort (and perhaps some more background knowledge too). Perhaps it would be useful to also cite the paper "Large deviations", Satya N. Majumdar, Gregory Schehr, arXiv:1711.07571 which is at a much more introductory level?

  • "if there is only one largest left eigenvalue" - this is a little slack, I would recommend being more precise here, i.e., explaining that the intention is "if the geometric multiplicity of the largest eigenvalue is 1", and perhaps to remind the reader that this means that the dimension of the corresponding eigenspace is 1.

  • I am a little confused by the notations n_s and n_c at the end of page 11 (and also later in the text). Are they actually the same thing? If not, the pseudo-code at the end of page 11 seems a little strange as n_c is given as input but not used, while n_s is used but not defined/given anywhere.

  • I think it would be useful to refer explicitly to the url [2] for the c code whenever the code is mentioned, e.g. at the top of page 12.

  • Could the author say something about the importance of computational power for using the methods described in the text. E.g., for a certain example, what quality of results would one expect to obtain on a single PC? What about parallelization? How does the smallest probability that can be realistically reached in a simulation scale with the computational power? I think this is important because some people in the large deviations community are under the (I believe mistaken) impression that without a strong cluster, they have no hope of running any useful simulations, and they therefore do not even attempt to do so.

  • To give a more complete picture, I think it would be useful to add a brief discussion of limitations of the method presented in these lecture notes, and mention some possible alternative methods (genetic algorithms etc) with appropriate references, somewhere towards the end of the paper.

  • Typos in the Conclusion: "hundres of decades" --> hundreds of decades, "chosing" --> choosing, "cahnhing" --> changing (?).

Recommendation

Ask for minor revision

---

## Round 1 · Referee Report · Anonymous (Referee 3) · 2025-2-5

Report

The manuscript provides a well-structured and insightful introduction to numerical techniques for studying large deviations.

I think that the choice of the Bernoulli process as a recurring example is particularly effective, serving as an intuitive entry point for students unfamiliar with the field. The author ensures that abstract principles are immediately applicable, by consistently anchoring theoretical discussions in concrete computational implementations. In my view, the manuscript stands out for its emphasis on practice, with the inclusion of coding references.

This, I recommend the publication of the lecture notes in their present form. Their pedagogical clarity makes them a very valuable resource for students and for any researchers aiming to acquire a conceptual understanding, and a hands-on experience in numerical large-deviation sampling.

I simply list below some optional modifications.

  • Small details:

. eq.(12) and in the equation just after in the text (top of p.7): parentheses (or distinct indices i and j) could help the reading

. after eq.(15): "Perron–Frobenius theorem" could be mentioned, perhaps?

. before eq.(22): it could be stated in words that the score S depends on the full history \underline y .

. p.17: "In contrast to real physical systems" is a bit vague (as there are physical systems in which some Laplace transforms with a Laplace paramter with no specific signs makes sense). Maybe instead the contrast should be done with the canonical ensemble of Thermodynamics?

. pages 25-27 lack a bit visible structure; I would optionally suggest items or subsubsections

  • Typos: p.3 and p.4: 10⁶ and 10⁶ --> 10⁶ and 10⁸ p.11: sytsem

Recommendation

Publish (surpasses expectations and criteria for this Journal; among top 10%)

---

## Round 2 · Author Response

Dear editors,

thank you for the detailed report of the referees. I am pleased that
they are very positive. I have considered all recommendations
for the next version of the manuscript. Below you find detailed answers
to all recommendations.

Thus, the manuscript should now suitable for acceptance.

Yours faithfully
Alexander Hartmann

---

## Round 2 · List of Changes



---

## Editorial Decision

published